# Hydrophobic, Thermal Shock-and-Corrosion-Resistant XSBR Latex-Modified Lightweight Class G Cement Composites in Geothermal Well Energy Storage Systems

**DOI:** 10.3390/ma16175792

**Published:** 2023-08-24

**Authors:** Toshifumi Sugama, Tatiana Pyatina

**Affiliations:** Brookhaven National Laboratory, Upton, NY 11973-5000, USA; sugama@bnl.gov

**Keywords:** lightweight cement, thermally insulating cement, reservoir thermal energy storage, geothermal cement, thermal-shock-resistant cement, hydrophobic cement, XSBR latex, Portland cement

## Abstract

Energy losses can be significantly reduced if thermally insulating cement is used for energy storage and recovery. The thermal conductivity (TC) of the currently used cement is between 1 and 1.2 W/mK. In this study we assessed the ability of polystyrene (PS)–polybutadiene (PB)–polyacrylic acid (PAA) terpolymer (cross-linked styrene–butadiene rubber, XSBR) latex to improve thermal insulating properties and thermal shock (TS) resistance of class G ordinary Portland cement (OPC) and fly ash cenosphere (FCSs) composites in the temperature range of 100–175 °C. The composites autoclaved at 100 °C were subjected to three cycles, one cycle: 175 °C heat → 25 °C water quenching). In hydrothermal and thermal (TS) environments at elevated temperatures in cement slurries the XSBR latex formed acrylic calcium complexes through acid–base reactions, and the number of such complexes increased at higher temperatures due to the XSBR degradation with formation of additional acrylic groups. As a result, these complexes offered the following five advanced properties to the OPC-based composites: (1) enhanced hydrophobicity; (2) decreased water-fillable porosity; (3) reduced TC for water-saturated composites; (4) minimized loss of compressive strength, Young’s modulus, and compressive fracture toughness after TS; and (5) abated pozzolanic activity of FCSs, which allowed FCSs to persist as thermal insulators under strongly alkaline conditions of cement slurries. Additionally, XSBR-modified slurries possessed improved workability and decreased slurry density due to the air-entraining effect of latex, which resulted in further improvement of thermal insulation performance of the modified composites.

## 1. Introduction

This work is the last publication in a series of three papers focused on the effort to design cement composites with very low thermal conductivity (TC) for reservoir thermal energy storage (RTES) and heat recovery wells. In such systems cement sheaths can encounter significant thermal stresses when hot fluid is pumped into the relatively cold reservoir or when hot fluid is recovered from it passing through the cool upper parts of the well. In the earlier two works we reported the formulation and testing of a highly hydrophobic lightweight cement composite with a TC of 0.28–0.5 W/mK under water-saturated conditions [1,2]. The first of the developed technologies involved the use of hollow fly ash cenospheres (FCSs) treated with polymethylhydrosiloxane (PMHS), that chemically tailors the microsphere surfaces making them hydrophobic and providing their stability under alkaline cement environments, in combination with calcium aluminate cement (CAC). This technology allowed decreasing the TC of the composite to 0.4 W/mK and was designed to be used in high-temperature RTES where the composite would set at 250 °C and experience a thermal gradient of 225 °C [1].

The second technology involved ultralight silica aerogel (SAG) treated with hexamethyldisilazane (HMDS). Reactions of the functional amine (=NH) group within HMDS with the silanol (≡Si-OH) group present on SAG surfaces initiated the trimethylsilylation of SAG surfaces forming trimethylsiloxy [−O-Si≡(CH_3_)_3_]-linked SAG, 2≡SiOH + (CH_3_)_3_≡Si–NH–Si≡(CH_3_)_3_ (HMDS) → 2≡Si–O–Si≡(CH_3_)_3_ + NH_3_↑, thereby, like for PMHS, leading to the presence of hydrophobic trimethylsilyl [−Si≡(CH_3_)_3_] groups on SAG surfaces in conjunction with ammonia release. Like for the FCSs treated with PMHS the hydrophobic surfaces of SAG prevented alkali dissolution of SAG and ingress of water. The treated SAG was incorporated into calcium aluminum phosphate cement slurry consisting of calcium aluminate cement, sodium hexametaphosphate. The porewater pH of that cement was 10.5. Like PMHS-treated FCSs, the hydrophobic behavior of cement was improved with a higher content of HMDS-treated SAG. Based upon the thermal degradation of HMDA at around ~250 °C, this composite that sets at 100 °C is resistant to TS at a thermal gradient of 150 °C and possesses a low TC of 0.28 W/mK.

On the other hand, in our work aimed at mitigating the brine-led corrosion of a carbon steel (CS) casing with an alkali-activated air-foamed lightweight calcium aluminate cement sheath, we evaluated the ability of an acrylic polymer emulsion containing functional carboxyl (-COOH) and alkyl ester (-COOR) groups to protect CS against corrosion. In the cement slurry, cations of calcium, sodium, and aluminum, and hydroxide ions, liberated by cement hydration and the activator, formed (COO^−^)_n−_-M^n+^] complexes (M: Ca, Na, and Al; n: cationic charges of M) with the polymer [3]. These derivatives not only offered an improved thermal stability of the polymer, but also served in impeding the permeation of corrosive electrolytes through the foamed cement, providing cathodic protection of CS against hot brine-caused corrosion. Additionally, in this study, we visually observed the water-repellent behavior of the polymer-modified cement surfaces.

Thus, in this paper we report the use of the cross-linked carboxylated styrene–butadiene rubber (XSBR) latex as terpolymer containing functional carboxyl groups for modifications of a class G OPC composite with FCSs and microglass fibers (MGFs) for applications in low (100 °C) temperature RTES systems. 

Compared with conventional non-carboxylated styrene–butadiene copolymer latex, XSBR latex has advantages in construction and oil fields. In construction of flexible pavement in cold regions, XSBR-modified asphalt provides both better high- and low-temperature performances, improved mechanical properties, and environmental hazard decrease by eliminating emissions of poisonous and pernicious gases due to the filling-effect of XSBR particles of tiny voids present in large surface area of asphalt [4]. In the studies of XSBR-modified OPC and of the reactions of carboxylate anions formed by deprotonation of carboxylic acid group in XSBR with calcium cations from cement, it was suggested that the absorption of anionic XSBR on hydrating cement acted to restrain the formation of ettringite, leading to a lack of reactive silica as well as suppression of portlandite precipitation, resulting in retardation of cement hydration reactions [5,6,7]. Also, in oil field cementing subjected to cyclic steam stimulation, similar effects of XSBR were observed for class G well cement; namely, the ettringite formation was delayed and decreased, and furthermore, an extension of the pre-induction and induction periods of cement hydration was observed [6,8]. On the other hand, the modification of XSBR with itaconic acid that includes two functional carboxylic acid groups offered adequate set control and improved flexural strength of well cement [7,9]. Thus, the same chelation mechanism of the carboxyl group with counter cations has been reported in other studies. 

XSBR latex is originally colloidal with numerous hydrophobic alkene chains and rings and hydrophilic functional carboxyl groups, which can react with hydrating cement forming chelate complexes. The coalescence of colloidal particles of XSBR chelated by a cement matrix results in the formation of a hydrophobic film on hydrated cement. Additionally, this hydrophobic continuous film may cover the surfaces of FCSs, protecting them from pozzolanic reactions in alkali cement. Hence, unlike in the case of the two technologies reported previously, where hydrophobic insulating composites were achieved by the inclusion of superhydrophobic PMHS- and HMDA-treated insulating aggregates [1,2], in this case the whole composite matrix including the insulating aggregates would be hydrophobic. Since the ingress of water into cement is undesirable for thermal insulation, the hydrophobic cement matrix may have better water-proofing performance than composites with hydrophobic aggregates. 

Another important geothermal-well cement function is protection of carbon steel (CS) casings against brine-caused corrosion for sustained well casing integrity. Decreased ingress of corrosive well fluids can enhance the corrosion-protection performance of cementitious composites. Various forms of latex have been successfully used for improved cement and steel casing protection from corrosion for applications in acidic underground wells [10,11,12,13,14,15,16].

In this paper the following factors governing the potential of XSBR latex-modified class G cement composites with FCSs as a thermal insulator and MGFs as a reinforcement for applications in RTES systems were investigated: (1) hydrothermal stability and oxidative degradations of XSBR latex at 100 °C and 175 °C after its solidification in an oven at 100 °C, (2) hydrothermal interactions mechanisms between colloidal hydrating cement grains and XSBR, (3) phase composition and transitions of XSBR-modified neat class G cement, (4) calorimetric studies of XSBR-modified neat cement slurry at 25 °C and 85 °C for assessing the changes in cement hydration behavior as a function of polymer/cement (P/C) ratio, (5) properties of XSBR latex-modified cement composite slurries including water/cement composite (W/C) ratio, density, and slump size, (6) protection of FCSs against pozzolanic reactions, (7) TS resistance of XSBR-modified cement composites based on the changes in their mechanical properties, (8) hydrophobicity of composite surfaces before and after TS test, (9) thermal conductivity of the composites before and after the TS tests, (10) microstructure exploration and characterization of freshly fractured composite surfaces, (11) effect of XSBR composites modification on the CS corrosion protection. 

## 2. Materials and Methods

### 2.1. Starting Materials

Class G well cement (OPC) was supplied by Trabits group, and the X-ray powder diffraction (XRD) data showed that its crystalline composition included four principal phases, hatrurite (ICDD# 04-014-9801, 3CaO.SiO_2_, C_3_S), calcio-olivine (ICDD#04-012-6734, CaO.SiO_2_, C_2_S), brownmillerite (#04-007-5261, 4CaO.Al_2_O_3_, Fe_2_O_3_), and calcium sulfate (#01-074-1905, CaSO_4_.2H_2_O). CenoStar Corp. (Newburyport, MA, USA) provided the fly ash cenospheres (FCSs) under the trade name “CenoStar ES500”. The FCSs’ bulk density is 0.32–0.45 g/cm^3^ and TC is 0.1–0.2 W/mK [1]. The major crystalline phases of FCSs were mullite (ICDD# 04-016-1586, Al_2_._22_ Si_0_._78_ O_4_._89_) and silica (#04-008-8437, SiO_2_). The cumulative size distribution of FCSs was as follows: 3 wt.% 300 µm, 54 wt.% 150 µm, 19.5 wt.% 106 µm, 15 wt.% 75 µm, and 8.5 wt.% < 74 µm. 

The non-crystalline micro-E-glass fibers (MGFs) with a size of 16 µm diameter and 120 µm length and a bulk density of 0.93 ± 0.08 g/cc under the trade name “Microglass 7280,” were obtained from Fibertec (Bridgewater, MA, USA). The surface of the fibers was treated with cationic agent for chemical protection. MGFs were used to improve the compressive fracture toughness to suppress and control the post-stress cracks’ opening and propagation. U.S. Silica Corporation provided us with silica flour with a particle size 40–250 µm. XSBR latex was supplied by Cudd Energy Services.

The AISI 1008 cold rolled steel test panels according to ASTM D 609C were supplied by ACT Test Panels, LLC (Hilldale, MI, USA). Alkaline cleaner #4429, supplied by American Chemical Products, (Cleveland, OH, USA), was used to remove surface contaminants of the CS. This cleaner was diluted with deionized water to prepare a 5 wt.% cleaning solution. 

Table 1 shows the oxide compositions of the starting materials.

### 2.2. Cement Formulas and Sample Preparation

The samples were prepared by dry blending OPC, FCSs, and silica flour. The XSBR latex was added to the mixing water prior mixing with the dry blend. The dry blend consisted of 38% OPC, 38% FCSs, 15% silica flour, and 9% MGFs by weight. To determine the content of the solid polymer (P) in XSBR latex, the latex was dried for 3 days in an oven at 100 °C until it reached a constant weight, the measured composition was ~43 wt.% P and ~57 wt.% water. Three different concentrations of XSBR latex were used for P/cement composite (C, dry blend) slurries: 5, 15, and 25 wt.%. The total water (W, added water + water in latex)/C ratios were 0.51, 0.45, 0.37, and 0.31 for 0, 5, 15, and 25% P/C formulations, respectively. The slurries were prepared by adding water with latex to the dry blend and mixing by hand for a minute. Slurries were poured into different molds for different types of tests and left for 24 h at room temperature. Then the hardened composites were demolded and placed in 99 ± 1% relative humidity for 24 h at 85 °C; finally, the composites were autoclaved in a non-stirred Parr Reactor 4622 (Hillsboro, OR, USA) for 24 h at 100 °C.

Also, in the following three studies, (1) the identification of the reaction products between neat OPC and XSBR, (2) the hydration behavior of XSBR-modified OPC, and (3) the hydrothermal stability and molecular alterations of XSBR in neat OPC after TS tests, XSBR-modified OPC samples with 0, 5, 15, and 25% P/C ratios and respective W/C ratios of 0.49, 0.48, 0.44, and 0.42, were prepared.

For CS corrosion tests, the samples were prepared in the following sequence. First, the CS panels (32 mm wide by 100 mm long, with 0.9 mm thickness) were immersed in a 5 wt.% alkali-cleaning solution at 40 °C for 10 min; second, cleaned panels’ surfaces were rinsed with tap water at 25 °C and dried for 24 h in air at room temperature; third, the panels were dipped into a bath with composite slurries at room temperature and withdrawn slowly; fourth, the composite slurry-covered panels with a covered area of 14,400 mm^2^ (45 mm long by 32 mm wide) were left for 24 h at room temperature, allowing the slurry layer to transform into a solid, followed by placing them in 99 ± 1% relative humidity with exposure for 24 h at 85 °C and, finally, the composite-coated panels were autoclaved for 24 h at 100 °C prior to conducting the electrochemical corrosion tests. 

### 2.3. Measurements

The slump size of the slurries, in mm, was measured using a 40 mm high polyethylene cone with dimensions of the hole at the top 20 mm in diam., and at the bottom 45 mm in diam. The cement slurry was filled in the cone placed on a flat carbon steel plate. Thereafter, the cone was slowly lifted, allowing the slurry to flow out. The diameter of the spread slurry, in mm, was measured 20 s later. 

For the TS tests the 100 °C-autoclaved samples were heated for 24 h in an oven at 175 °C and then immersed into 25 °C water for 15 min. This heat-quenching process was repeated three times. The extent of TS resistance was evaluated from the changes in mechanical properties after TS. To evaluate hydrothermal stability and molecular alterations of dry-solidified XSBR it was autoclaved at 100 and 175 °C for 24 h.

Attenuated total reflectance-Fourier transform infrared spectroscopy (ATR-FTIR, Perkin Elmer Spectrum 100, Waltham, MA, USA), TGA/DTA (heating rate of 20 °C/min in a N_2_ flow, model Q50, TA Instruments, New Castle, DE, USA), and X-ray diffraction measurements (40 kV, 40 mA copper anode X-ray tube, Rigaku Smartlab, Cedar Park, TX, USA) were used for XSBR sample characterization and for sample analyses before after the TS tests. The PDF-4/Minerals 2022 database of International Center for Diffraction Data (ICDD) was used for analyses of XRD patterns.

To assess water-repellency of cured composites, contact angle measurements for a water droplet on air-dried composite surfaces were performed with Model CAA 3, Imass Inc., using rectangular prism samples (15 × 75 × 2 mm). Prior to these measurements, all samples were left open to the air at room temperature for 7 days to prepare air-dried surfaces. Each contact angle value is an average of the measurements performed at five different locations.

To evaluate composites’ water-fillable porosity of cylindrical samples (20 mm diam. × 40 mm height) after curing in a 100 °C autoclave was computed using W_wet_ − W_dry_/V_wet_ × 100, where W_wet_ is the weight of water-saturated samples and W_dry_ is the weight of samples dried for at least 4 days in a vacuuming oven at 65 °C until the weight of the sample did not change anymore, V_wet_ is the volume of the sample. 

To obtain information on hydration of class G composite slurries with different P/C ratios a calorimetric study was performed at the isothermal temperatures of 25 °C and 85 °C, using a TAM Air Isothermal Microcalorimetry (TA Instruments, New Castle, DE, USA). In this study the initial and final setting times and the heat energy evolved during the acid–base and hydration reactions were determined. 

For the compressive strength and compressive toughness, the composite samples were prepared in cylindrical molds (20 mm diam. and 40 mm height). An electromechanical Instron System Model 5967 (Norwood, MA, USA) was used to obtain these mechanical properties. Compressive strength is the capacity of a material or structure to resist or withstand compression. On the other hand, cement with a high compressive strength may be brittle and lack stress energy absorption, which results in a rapid propagation of pre-existing and newly created cracks in cement bodies under stress conditions. Thus, an ideal cement is required to possess high toughness properties that delay or totally prevent crack propagation. To obtain quantitative data of compressive toughness, we determined the total energy consumed during the completion of cement’s compressive failure; it was computed from the enclosed area of the compressive stress–strain curve with the baseline extending between the beginning and the end of the curve. The toughness depends primarily on the balance between compressive strength and ductility.

JEOL 7600 F (Pleasanton, CA, USA) scanning electron microscope image analysis coupled with energy dispersive X-ray (EDX) elemental composition measurements on freshly broken cement samples was employed for morphological analyses and phase identifications. 

The TC was measured using a Quick Thermal Conductivity Meter, OTM-500, Kyoto Electronic on rectangular prism samples (60 × 120 × 20 mm) using a water-proofed probe consisting of a single heater and thermocouple. The probe was placed on the cement surface after extra water was removed from the water-saturated cement surface with a paper towel. 

To obtain information on CS protection by modified composites against brine-caused corrosion, DC electrochemical testing of CS plates covered with the composite was performed using a Princeton Applied Research Model Versa STAT 4 Corrosion Measurement System. In this assessment, the P/C 0, 5, 15, 25 ratios of cement composites before and after TS tests were mounted in a holder, and then inserted into an Ametex Model K0235 flat cell containing a 1.0 M sodium chloride electrolyte solution. The test was conducted under an aerated condition at 25 °C, on an exposed working electrode surface area of 1.0 cm^2^. The polarization curves were measured at a scan rate of 0.17 mVs^−1^ in a corrosion potential range from −0.4 to +0.6 V. The average corrosion rate, in mm/year, associated with the corrosion potential E_corr_. (V) and the corrosion current density, I_corr_. (A) was obtained from Tafel plot fit results of polarization curves at three different locations for the cement-coated CS plates. The thickness of the coating was estimated using an Absolute Digimatic Caliper from Mitutoyo Corp. 

## 3. Results and Discussion

### 3.1. Hydrothermal Stability of XSBR

To investigate the hydrothermal stability of solid XSBR, its film sample was prepared by drying “as-received” latex for 24 h in an oven at 100 °C. The film was then autoclaved for 24 h at 100 °C and 175 °C, respectively. 

Figure 1 shows the ATR-FTIR spectra of the 100 °C-dried, 100 °C-autoclaved, and 175 °C-autoclaved film samples. Firstly, we describe styrene-related bands and then butadiene-related ones. For the 100 °C-dried sample, since the bands in the range of 3055–3020 cm^−1^ belong to alkene (C=C) groups in aromatics [17,18], the absorption bands at 3085, 3061, and 3026 cm^−1^ are attributable to stretching vibration (ν_C=C_) of cyclic alkene (C=CH-)_n_ in aromatic groups, while the bands at 1492, 1110, and 697 cm^−1^ are due to the C-C bond stretching (ν_C-C_), in-plane C-H bending (δ_C-H_), and out-of-plane C-H bending (δ_C-H_) in aromatic groups, respectively [17,19]. Regarding alkane (-CH_2_-) chain and (-CH_3_) end groups, there are seven relevant bands at 2993, 2913, 2893, 2865, 1452, 1346, and 758 cm^−1^. The first two bands are implicated in C-H asymmetric (ν_as C-H_) stretching of CH_3_ and CH_2_, respectively; correspondingly, the band at 2893 and 2865 is the C-H symmetric (ν_s C-H_) mode of CH_3_ and CH_2_ [19]. The bands at 1452, 1346, and 758 cm^−1^, are assignable to ν_C-H_ in CH_3_ [19,20], and the scissors and rock of C-CH_2_ bonds in chain alkanes, respectively [17,19,21].

As for the polybutadiene (PB)-related bands, the alkene in diene of PB appears in a very narrow range of 3100 to 3000 cm^−1^ [18]. Thus, the absorption frequency at 2993 cm^−1^ was not only related to the alkane CH_3_, but also was overlapping with ν_C=C_ of diene alkene. The other ν_C=C_ alkenes can be seen as the band at 1600 cm^−1^. On the other hand, the contributors of 966, 908, and 803 cm^−1^ bands are *trans*-1,4 units, 1,2 vinyl units, and *cis*-1,4 units, respectively [20,22,23]. As to the carboxyl group of XSBR, the O-H stretching vibration (ν_O-H_) in -COOH also exists in the range of 3500–2500 cm^−1^ [24], reflecting that a very broad band from 3552 to 3225 cm^−1^ involves COOH-related ν_O-H_. Furthermore, the bands at 1730 and 1247 cm^−1^ are ascribable to C=O stretching vibration (ν_C=O_) and C-O stretching (ν_C-O_) modes, respectively [24,25,26]. 

As a result, since this styrene–butadiene copolymer contains the functional carboxyl group, an additional polyacrylic acid polymer is likely to be incorporated into this copolymer. Thus, the solid-state conversion of “as-received” XSBR latex can be drawn as the terpolymer structure shown in Figure 1 [27]. 

These dried samples next were autoclaved for 24 h at 100 °C and 175 °C, followed by an additional drying in an oven at 100 °C. As seen in Figure 1, the spectral feature of the 175 °C-autoclaved sample differed from the pre-autoclaved sample, while no noticeable difference was observed for the 100 °C-autoclaved sample. For the 175 °C-autoclaved sample there were two differences: the enhancement in absorbance height, ΔA, of COOH-related ν_C=O_ and ν_C-O_ bands at 1730 and 1247 cm^−1^; and the decrease in the ΔA value of PB diene alkene-led bands including ν_C=C_, trans-1,4, 1,2 vinyl, and *cis*-1,4 units at 1600, 966, and 803 cm^−1^. This fact strongly suggested that PB diene was susceptible to hydrothermal degradation and oxidation at 175 °C, resulting in transformation into COOH. To obtain the quantitative information on such alteration, the height ratio (ΔA_1730 cm−1_/ΔA_1600 cm−1_) between -COOH at 1730 cm^−1^ and PB C=C at 1600 cm^−1^ at 100 °C and 175 °C was computed (Table 2). The height ratio of 0.471 for 100 °C-dried XSBR latex was enhanced by nearly 4.5 times to 2.1 after being autoclaved at 175 °C; correspondingly, both values of ΔA_966 cm−1_ of PB *trans*-1,4 and ΔA_908 cm−1_ of PB 1,2 vinyl decreased by 0.03 and 0.017, respectively, from 0.125 and 0.034 of the pre-autoclaved one. 

Figure 2 depicts two-step hydrothermal degradation and oxidation pathways of PB *trans-* and *cis*-1,4, and 1,2 vinyl units [28,29,30].

The first step is the cleavage of diene alkene. The second step is the incorporation of carboxyl groups with formation of carboxyl (COOH)-terminated PB (CTPB) in the isolated conformation. Furthermore, the CTPB formation causes the alteration of PS and PAA to the carboxyl-ended PS and PAA conformations. If this pathway is rational, additional functional carboxyl groups appear to be incorporated into the degraded XSBR. In contrast, the aromatic alkene of PS is unsusceptible to hydrothermal oxidation. In fact, there were no significant changes in the ΔA values of aromatic alkene-related bands at 1492, 1110, and 697 cm^−1^. 

### 3.2. Products of Hydrothermal Reaction between XSBR and OPC

Figure 3 shows ATR-FTIR spectra of 100 °C- and 175 °C-autoclaved neat OPC. Their spectral features are very similar, suggesting that the hydration products of OPC cured in a hydrothermal temperature range of 100 °C to 175 °C for 24 h are almost the same. The spectra encompass eight representative bands. The band at 3645 cm^−1^ is relevant to the O-H bond stretching (ν_O-H_) vibration in portlandite [Ca(OH)_2_, CH] [31,32]. Water (H_2_O) molecule-related bands can be seen from O-H symmetric stretching vibration (ν_s O-H_) at broad 3345 cm^−1^ band and H-O-H bending vibration (δ_H-O-H_) at 1640 cm^−1^ [33]. The three bands at 1480, 1426, and 872 cm^−1^ are associated with calcite (CaCO_3_)-related carbonate CO_3_^2−^; namely, C-O asymmetric stretching (ν_as C-O_) at 1480 and 1426 cm^−1^, and a O-C-O out-of-plane bending (δ_O-C-O_) mode at 872 cm^−1^ [34,35,36]. The shoulder band at 1132 cm^−1^ is attributed to S-O symmetric stretching (ν_s S-O_) in the sulfate, SO_4_^2−^, of gypsum [31,37]. The prominent peak at 960 cm^−1^ is calcium silicate hydrate (CaO-SiO_2_-H_2_O, C-S-H)-related Si-O asymmetric stretching (ν_as Si-O_) in silicate, SiO_4_^4−^ [31,38,39]. 

Figure 4 represents ATR-FTIR spectra of XSBR-modified OPC after autoclaving at 100 °C and 175 °C. For the 100 °C-autoclaved XSBR-OPC sample comparison with the XSBR spectrum (Figure 1) and OPC spectrum (Figure 3) alone, the following differences appeared: (1) the appearance of bands at 1592 and 1411 cm^−1^; (2) a striking decay of CO_3_^2−^ -related peak intensity at 872 cm^−1^, and disappearance of other CO_3_^2−^ bands at 1480 and 1426 cm^−1^; (3) the absence of PB *trans*-1,4, 1,2 vinyl unit and *cis*-1,4 bands at 966, 908, and 803 cm^−1^; and (4) there were no COOH-related ν_C=O_ and ν_C-O_ bands at 1730 and 1247 cm^−1^. Regarding the changes (2) and (4), since in neat OPC calcium carbonate forms in acid–base reactions of Ca^2+^ from cement dissolution and carbonate/bicarbonate ions, the fact that its formation is suppressed suggests that Ca^2+^ and OH^−^ from hydrolyzed OPC preferentially react with PAA- carboxyl groups (proton donor carboxylic acid, -COOH^+^) within XSBR through acid–base reaction, 2-COOH^+^ + Ca^2+^ + 2OH^−^ → 2-COO^−^ − Ca^2+^ (chelation) + 2H_2_O. This is the reason why no COOH-related bands and considerable carbonation reduction were observed in XSBR-modified cement samples. Furthermore, such acid–base reaction resulted in the alteration of functional carboxyl groups into negatively charged carboxylate anion (-COO^−^) chelating calcium ions. The presence of this COO^−^ can be identified by the asymmetric and symmetric stretching bands emerging in two frequency ranges, 1650–1540 and 1450–1360 cm^−1^ [40]. Thus, returning to the first change: the peak at 1592 cm^−1^ and shoulder band at 1411 cm^−1^ were due to the COO^−^ asymmetric (ν_as COO−_) and symmetric stretching (ν_s COO−_) modes, respectively, in this chelated Ca complex [3,41]. The alkene- and alkane-associated bands were present in the 3085–3026 and 2993–2865 range, respectively, in conjunction with 1452, 1346, and 758 cm^−1^; furthermore, aromatic bands were visible at 1492, 1082, and 697 cm^−1^. The cement hydration product (C-S-H)-related band at 960 cm^−1^ remains one of the major peaks. 

Compared with at 100 °C, at 175 °C, three differences can be pointed out: the emergence of 1730 and 1241 cm^−1^ bands; a 2.2-times increase in the ΔA_1592 cm−1_ value of 0.0035 at 1592 cm^−1^ compared to the 100 °C-autoclaved sample; and the disappearance of all CO_3_^2−^-related bands. Regarding the first difference, as mentioned earlier, the diene alkene in PB was susceptible to the hydrothermal degradation/oxidation at 175 °C resulting in its alteration into isolated CTPB, and non-isolated CEPS and CEPAA. Thus, CTPB, CEPS, and CEPAA would be transformed into Ca^2+^-coordinated PB major complexes, and CEPS and CEPAA into minor complexes in the presence of OPC. Since the pendant carboxyl groups in PAA also alter to carboxylate anions, PAA has two Ca^2+^-coordinated chelate complexes at the end and as a pendant. 

Based on the above information, Figure 5 illustrates molecular transformations of XSBR in the presence of Ca^2+^ at hydrothermal temperatures of 100 °C and 175 °C. Figure 6 illustrates two types of chelate complex configurations derived by hydrothermal degradation/oxidation of PB at 175 °C. The first type was a PAA pendant complex formed at 100 °C; the second type were the multiple complexes formed at 175 °C. This means that although XSBR was partially degraded at 175 °C, this elevated temperature yielded more functional carboxylate anion groups sequestering Ca with the formation of additional -COO^−^—Ca^2+^ chelate complexes. This extensive complexation eliminated formation of CaCO_3_ at 175 °C.

### 3.3. Crystalline Phase Composition

The XRD patterns of crystalline phases of 175 °C-autoclaved XSBR-modified and neat OPC are shown in Figure 7. For the neat OPC (red, bottom XRD pattern), three crystalline phases were identified: portlandite (CH) and CaCO_3_ as major phases and 9 Å tobermorite as a minor one. Relating this information to the ATR-FTIR results, the C-S-H phases identified in FTIR may include 9 Å tobermorite along with the amorphous C-S-H. In contrast, the XRD patten (blue, top) of modified OPC showed four main differences: (1) elimination of CaCO_3_ peaks; (2) presence of three unreacted OPC phases, hatrurite (3CaO.SiO_2_, C_3_S), calcio olivine (2CaO.SiO_2_, C_2_S), and brownmillerite [Ca_2_(Al,Fe)_2_O_5_, CAF]; (3) absence of 9 Å tobermorite; and (4) appearance of katoite [Ca_3_Al_2_(SiO_4_)_3−x_(OH)_4x_, C-A-S] as a new minor phase, while CH was still present as a major phase. Regarding the first difference, the disappearance of CaCO_3_ agrees with the FTIR results that suggested interactions of carboxyl groups in three PB degradation/oxidation derivatives, CTPB, CEPS, and CEPAA, with Ca and inhibition of formation of calcium carbonate, but interestingly, CH still forms as a result of OPC hydration. This fact may come from a rapid crystallization of CH despite Ca-sequestration activity of these derivatives. On the other hand, it is well known that CH is very susceptible to carbonation with CO_2_ in air and CO_3_^2−^ ions in water [42,43,44]. For the presence of CH as the major phase at 175 °C, a possible explication is its protection from carbonation by formation of type 2 multiple chelate complexes on hydrated OPC grain surfaces [5,45]. Likewise, the second difference is due to a possible adsorption of type 2 complexes on OPC surfaces, therefore delaying the hydration reactions of C_3_S, C_2_S, and CAF. FTIR showed the presence of C-S-H, which was not visible in XRD patterns. Therefore, this C-S-H was essentially amorphous. Although not shown, similar patterns were obtained for 100 °C-autoclaved XSBR-modified and neat OPC samples. 

### 3.4. Thermal Stability of XSBR-OPC Reaction Products

To obtain information on thermal stability of XSBR-modified OPC, two reference samples for comparison were prepared. One was the 100 °C-dried XSBR sample (no cement) followed by 100 °C- and 175 °C-autoclaving; the other was 100 °C- and 175 °C-autoclaved neat OPC. Figure 8 depicts the TGA/DTG curves of XSBR reference samples. For the 100 °C-dried XSBR sample, the decomposition onset temperature was T_o_, 250 °C, the maximum decomposition temperature was T_md_, 422 °C, and the decomposition end temperature was T_e_, 493 °C. Thus, the 100 °C-dried bulk XSBR is likely to withstand temperatures up to ~250 °C. Also, a total mass loss of 97.5% was detected from thermal decomposition of XSBR; correspondingly, the carbon release rate from degraded XSBR obtained using the area under the curve was 4.82%.min/°C. Similar T_o_, T_md_, total mass loss, and carbon release rates were observed for 100 °C- and 175 °C-autoclaved XSBR samples. Since hydrothermal decomposition/oxidation of PB within XSBR at 175 °C led to the formation of type 2 multiple chelate complexes as oxidative derivatives, these derivatives appear to be stable at temperatures up to ~250 °C. 

Figure 9 presents TGA-DTG curves of a 100 °C-autoclaved OPC reference sample showing three main decomposition steps [46,47,48]. The initial decomposition mass loss in a temperature range between 25 °C and 254 °C corresponds to the dehydration of amorphous C-S-H. This decomposition is followed by the dehydroxylation of CH and 9 Å tobermorite between 254 °C and 522 °C. The last mass loss is decarbonation in a 555–697 °C temperature range. As to the dihydroxylation step, since 9 Å tobermorite was formed as a minor phase, this decomposition mainly belongs to CH. The mass losses of C-S-H, CH, and CaCO_3_ were 4.2, 7.2, and 1.7%, respectively. For CH [Ca(OH)_2_], the maximum decomposition occurs at 451 °C (T_md_) according to the theoretical dehydroxylation reaction, Ca(OH)_2_ → CaO + H_2_O. The dehydroxylation rate computed using molar mass loss of H_2_O, (H_2_O/Ca(OH)_2_) × 100 = H_2_O%, was 24.3% for 1 mole Ca(OH)_2_. Thus, 7.2% loss in this study corresponds to 0.296 mole Ca(OH)_2_, which is equivalent to ~21.9 g Ca(OH)_2_ present in 100 °C-autoclaved cement. The computed decarbonation of CaCO_3_, CaCO_3_ → CaO + CO_2_, leads to a theoretical mass loss of 44% CO_2_ per 1 mole CaCO_3_. Based on this, the 1.7% CO_2_ molar mass losses imply that cement carbonation resulted in ~3.9 g of CaCO_3_. Increasing hydrothermal temperature to 175 °C engendered more carbonation of OPC. The CO_2_ molar mass loss increased by nearly 2-fold to 3.2%, compared with that of 100 °C sample. Thus, ~7.0 g CaCO_3_ was formed in this OPC. In contrast, the dehydroxylation rate of Ca(OH)_2_ declined to 6.8% from 7.2% of 100 °C OPC. This rate decline is equivalent to the decrease of ~20.7 g Ca(OH)_2_, while the quantity of C-S-H remained unchanged.

Figure 10 shows the TGA-DTG curves of 100 °C- and 175 °C-autoclaved XSBR/OPC samples. The curve of the 100 °C-autoclaved blend clearly had the combined features of both XSBR and OPC curves; namely, there were three different mass-loss events. The first one with a 2.1% mass loss, was from C-S-H; the second one with a large mass loss of 64.6% in the range of 228–502 °C was likely associated with XSBR in conjunction with some Ca(OH)_2_, and the final stage was decarbonation of 4.5% from CaCO_3_. The small decarbonation mass loss of the 175 °C-autoclaved XSBR/OPC blend strongly supported the FT-IR and XRD results. As explained above, the two factors that prevented cement carbonation were the uptake of Ca^2+^ by CTPB, CEPS, and CEPAA and the adsorption of type 2 complexes on hydrated OPC grain surfaces to impede carbonation of CH at 175 °C. 

### 3.5. Changes in Hydration Behavior of OPC by XSBR

Figure 11 gives the isothermal calorimetry curves for up to ~52 h and the exothermic hydrothermal reaction energy (EHRE, J/g) computed using the integration of normalized heat flow curves for OPC slurries made with 0, 5, and 15% P/C ratios. The curves indicate two-stage hydration reactions. The first stage reactions occurred within an elapsed time of less than ~2 h from the beginning of the signal registration; for the second stage, the onset of the hydration took place in the range of ~3 h to ~8 h. According to the literature [6,49,50], in the induction period of OPC hydration within the first 2 h (stage I), a primary hydration product is ettringite (Ca_6_Al_2_(SO_4_)_3_(OH)_12_·26H_2_O) derived from the reactions between the hydrolysates of hydrolyzed C_3_A and gypsum (CaSO_4_·2H_2_O). C_3_S and C_2_S hydrolyze forming C-S-H and CH during stage II. The value of EHRE during stage I was 3.39 J/g for non-modified OPC. At 5% P/C, the generated 3.13 J/g was ~8% lower than that of 0% P/C, suggesting that the quantity of ettringite was somewhat reduced in the presence of XSBR. A striking reduction in EHRE down to 1.77 J/g was observed for the 15% P/C ratio sample, implying that the increased concentration of XSBR restrained ettringite formation. This result is in agreement with the data reported by previous investigators [6,45]. Although this information was obtained at ambient temperature, type 1 PAA pendant chelate complexes may be formed by acid–base reactions between PAA carboxyl groups as proton donors and Ca^2+^, 2OH^−^ (proton acceptor) from C_3_A and gypsum, followed by possible adsorption of type I complexes on OPC grain surfaces as illustrated in Figure 11. 

The stage II is attributable to the hydration reactions of C_3_S and C_2_S, leading to the formations of C-H-S and CH. The onset hydration time appeared to be longer for higher P/C ratio samples. A 3 h:28 min of 0% P/C extended to 7 h:40 min with the 15% P/C. Thus, two factors, the uptake of Ca^2+^ liberated from hydrolyzed C_3_S and C_2_S by PAA carboxyl and the adsorption of type I complexes formed by this uptake on OPC, delayed the hydration reactions of stage II. Assuming 57 h as the end time for all curves, the computed EHRE values were 96.8, 83.6, and 72.5 J/g for the 0%, 5%, and 15% P/C ratios, respectively. The prolonged onset hydration time corresponded to a lower EHRE. Not surprisingly, the EHRE generated during both C-H-S and CH formations was considerably higher than EHRE of the ettringite formation. 

At an elevated temperature of 85 °C (Figure 12), the calorimetry curves disclosed just a single heat release event. Since ettringite decomposes at 90 °C [51] it may decompose shortly after the formation at the test temperature. On the other hand, the EHRE and onset of heat release time data show that the elevated temperature accelerates cement hydration reactions with formation of C-S-H and CH. The EHRE values obtained for the 0% and 15% P/C ratio samples were 146.7 and 105.5 J/g, respectively, which are ~46 and ~42% higher than that of total EHRE of stages I and II at 25 °C. For the latter, the reaction onset times of the 0% and 15% P/C ratios were 1 h:58 min and 5 h:17 min, respectively, corresponding to 1 h:30 min and 3 h:07 min shorter than that at 25 °C. As expected, the incorporation of more XSBR engendered the decline of EHRE values and prolonged the heat release onset time. As seen in the figure, the EHRE value of the 0% P/C was reduced by nearly 5% to 138.9 J/g for the 5% P/C ratio. A further reduction to 105.5 J/g was observed for the 15% P/C ratio sample; there was no significant change in EHRE for the 25% P/C ratio sample. Regarding the onset of heat release time, 5 h:17 min of the 25% P/C was more than 3 h longer than that of the 0% P/C ratio. The heat release time ranged from 8 h:46 min to 9 h:35 min for the tested formulations. The results described above strongly supported the following two factors, the uptake of Ca^2+^ by PAA pendant carboxyl and adsorption of type 1 complexes on OPC grain surfaces. Those factors not only contributed to inhibiting the carbonation of free Ca^2+^ and CH but were also effective in retarding the onset of cement hydration reactions. 

### 3.6. Slurry Properties

The densities and slump size of the slurries with different polymer-to-composite ratios are given in Table 3. XSBR has a strong dispersing effect on the OPC slurry. Even with the decreased W/C ratio from 0.51 for the neat slurry to 0.31 for 25% XSBR the size of the slump increased from 60 to 75 mm due to increased slurry fluidity. Another XSBR effect is decrease in the slurry density due to the air entraining in the presence of the polymer. The slurry density dropped from 1.26 to 1.04 for slurries with 0 and 25% XSBR, respectively. Note that an antifoaming agent was not used in the blends.

The data show that XSBR improves the workability/pumpability of the slurry.

### 3.7. Crystalline Phase Compositions

The crystalline phase compositions of 100 °C-autoclaved OPC-based composites with and without XSBR are shown in Figure 13. Peaks of the same crystalline phases can be found in both patterns for the most part. The patterns differ in the intensity of (1) portlandite peaks, (2) carbonate peaks, and (3) peaks of non-reacted cement phases. The higher intensity of the peaks from the original crystalline OPC phases (calcio-olivine and brownmillerite) in the sample with 25% XSBR latex suggests that the polymer slowed down cement hydration. As a result, there was still gypsum present in XSBR latex-modified cement but not in the neat sample. Latex also prevented the sample’s carbonation—peaks of carbonated phases are significantly lower in latex-modified cement than in the control. The results were similar for the slurries autoclaved at 175 °C (Figure 14). The detection of non-hydrated OPC phases is a result of the adsorption of type 1 PAA pendant complexes and type II multiple complexes on OPC at 100 °C and 175 °C, respectively, and slowing of cement hydration. There were no peaks of crystalline calcium-silicate hydrates after both curing temperatures, which means that they mostly remain amorphous after the short-time curing of 24 h. 

### 3.8. Molecular Alterations of XSBR in OPC Composites after Thermal Shock (TS) Tests

Figure 15 depicts ATR-FTIR spectra in a frequency range of 2000 to 650 cm^−1^ for pre-and post-TS modified, and unmodified cement composites autoclaved at 100 °C. The pre-TS unmodified composites displayed the peaks relevant to CaCO_3_ and C-S-H, whereas post-TS sample showed intensive signals of carbonation bands, strongly demonstrating that TS treatment promoted the carbonation of cement. The carbonation rates for both of 25% P/C-modified composites before and after TS were very low, if any. This carbonation-protection of pre-TS modified composite was produced by type 1 complexes and their adsorption on OPC surfaces. For the post-TS modified composite, the ΔA of complex-related bands at 1592 and 1411 cm^−1^ increased, indicating that TS tests yielded more functional carboxyls in 175 °C heat → 25 °C water quenching cycles. This fact was similar to the information obtained for 175 °C-autoclaved XSBR/neat OPC systems. Such additional carboxyl groups produced multiple type 2 complexes. This is the reason why, although the carbonation of OPC rose in TS tests, the multiple type 2 complexes and their adsorption on OPC essentially abated the carbonation of OPC for XSBR-modified samples.

### 3.9. Water Repellency of Composite Surfaces

The water-repellency of 100 °C-autoclaved composite surfaces dried in air for 7 days at ambient temperature before and after TS tests was evaluated by measuring the degree (°) of contact angle, θ, of water droplets over the composite surfaces. Depending on the water droplet contact angle the surface was categorized as superhydrophobic for >150°, over-hydrophobic in the range of 150° to >120°, hydrophobic in the range from 120° to 90°, and hydrophilic at <90° [52]. Figure 16 shows the changes in θ as a function of the P/C ratio for pre- and post-TS test samples in conjunction with the visual images of water droplets over the pre-TS test samples. For pre-TS samples, the surface of the 0% P/C ratio sample was hydrophilic with the water droplet spreading over the surface. Some improvement of water repellency was observed for the composite made with the 5% P/C ratio corresponding to θ of 36.8°. However, it was not enough to achieve hydrophobic properties. Increasing the P/C ratio to 15% made the surface hydrophobic, corresponding to the value of θ enhanced by nearly 1.7-fold to 100.3° compared with that of the 5% P/C. Further increasing the P/C ratio to 25% led to the achievement of an over-hydrophobic surface > 120°. For post-TS samples, all samples except for the 0% P/C improved the water repellency; for instance, the θ vale of the 5% P/C showed an enhancement of more than 2-fold over the pre-TS sample. Hence, type 2 complexes played an important role in improving hydrophobic properties in terms of excellent water repellency.

### 3.10. Water-Fillable Porosity of Composites

Water-fillable porosity (WFP) as an indicator of water-proofing of 100 °C-autoclaved water-saturated samples was evaluated before and after TS (Figure 17). The ideal composite with low WFP has a better waterproofing performance than a composite with high WFP. For pre-TS samples, a downward trend of WFP with increasing P/C ratio can be seen. Namely, the porosity of 38.9% for the 0% P/C was reduced by ~30% to 27.3% for the 25% P/C. After TS, the WFP of all samples declined, particularly the 15 and 25% P/C samples exhibited WFP reductions of >50%. Such porosity reduction could be attributed to the assemblage of a waterproofing-composite structure by type 2 complexes. 

### 3.11. Thermal Conductivity

Figure 18 shows the changes in thermal conductivity, λ, as a function of the P/C ratio for pre- and post-TS samples. For pre-TS samples, the λ (0.42 W/mK) of the 0% P/C decreased with increasing P/C ratio to 0.39, 0.35, and 0.32 for the 5, 15, and 25% P/C ratios, respectively. This result was correlated directly to three properties described above; (1) water repellence, (2) waterproofing, and (3) low slurry density induced by the air-entraining effect of XSBR. The high hydrophobicity associated with the No. 1 property, the minimum water ingress relevant to No. 2, and the insulating air bubble incorporation for No. 3 were responsible for reducing the λ value of water-saturated composites. The post-TS samples showed further considerable decrease in λ. For instance, the 15 and 25% P/C ratio samples with 0.25 and 0.23 W/mK thermal conductivity, respectively, were ~29 and ~28% lower compared with that of pre-TS samples made with the same P/C ratios. By contrast, thermal conductivity of the 0% P/C ratio sample declined only by 7% after the TS tests. Thus, formation of more type 2 complexes in composite resulted in a decreased thermal conductivity, improved water repellence, and water proofing.

### 3.12. Mechanical Properties

Figure 19 shows the compressive strength and the rate of its decline in TS tests for composites with different P/C ratios. Increase in latex concentration resulted in stronger cements. Before the TS, the composite made with the 25% P/C had a compressive strength of 16.6 MPa, which was ~75% higher than that of the control. After the TS, all composites lost some strength. The control composite lost 26.8% in strength after the TS. The rate of the strength reduction for XSBR-modified composites depended on the P/C ratio. The increase in the P/C ratio led to a lower strength decline in TS tests. The 5, 15, and 25% P/C ratio samples displayed losses of 20.2, 16.5, and 14.4% of the strength, respectively. Compared with the control, the 15 and 25% P/C composites experienced ~38 and ~46% lower strength losses. The data demonstrate that XSBR clearly improves composites’ resistance to TS.

The Young’s modulus (YM) data mirrored those for the compressive strength (Figure 20) with the modulus increasing at higher latex concentrations. A YM value of 859 MPa for the control increased by ~37% to 1176 MPa for the 25% P/C. After TS, the decline in YM ranged from ~24% for the control to ~13% for the 25% P/C compared with the values before the TS. The composites with high latex contents exhibited lower decline rates in stiffness. The loss in stiffness for all composites implied that the composites underwent ductile to soft transition during the TS tests. The 25% P/C composite sustained its ductility with minimal transition to softness. 

The compressive fracture toughness (Figure 21) increased with the increase in latex concentration. The 0.39 N-mm/mm^3^ toughness of the 0% P/C was enhanced by ~2.4-fold to 0.92 N-mm/mm^3^ for the composite made with the 25% P/C ratio. Since the toughness refers to the combination of ultimate strength and ductility, the addition of XSBR improved this combination. Like the strength and YM trends, all samples lost some of the toughness in TS tests. The toughness decline rate decreased with increasing P/C ratio. The 0, 5, 15, and 25% P/C ratios corresponded to ~46, ~33, ~26, and ~23% toughness decline rates, respectively. 

The data on mechanical properties of tested composites demonstrate that addition of high concentrations of XSBR latex improves strength and ductility of the composites and their TS resistance under the test conditions.

Figure 22 illustrates compressive fracture toughness measurements and comparison between features of compressive stress (MPa)–compressive strain (%) curves for the 0% and 25% PC composite samples after TS. As is shown in the figure, fracture toughness was computed from the sum of areas related to the ultimate strength (area A) and ductility (area B). Thus, an ideal compressive fracture toughness comes from a good combination of ultimate strength and ductility. The ultimate strength of the 0% P/C was reached at 8.14 MP compressive stress (Point 1). This stress initiated crack development. Thereafter, the compressive stress fell sharply to 2.38 MPa (Point 2), while the displacement extension of Point 1 (1.88%) to Point 2 (2.24%) was very short, only 0.36%. Beyond Point 2, the curve leveled off until displacement reached the end of the curve at failure (point 3). There were three different fracture toughness characteristics for the 25% P/C sample. Firstly, the stress reduction from Point 1 at ultimate strength to Point 2 occurred more gradually than in the case of the 0% P/C. Secondly, the displacement from Point 1 (2.14%) to Point 2 (4.32%) corresponded to a nearly 2-fold extension. Thirdly, the maximum displacement at Point 3, for the 25% P/C was ~24% longer than for the 0% P/C. From these considerations, it was obvious that the 25% P/C had a far better ductility than the 0% P/C composite. The MGF reinforcement might have provided some improvement in ductility hence in fracture toughness by delaying the crack propagation and suppressing the post-stress cracks’ initiation, opening, and crack width, in the case of the 0% P/C composites. However, the effectiveness of XSBR derivatives formed in TS in improving ductility was much greater than that of the limited MGFs in this work. Furthermore, oxidized XSBR derivatives offered development of high ultimate strength to the composite. Thus, such a great combination of high ultimate strength and improved ductility essentially led to the excellent toughness. 

The dual effect of MGF reinforcement and XSBR on compressive fracture toughness was evaluated on 100 °C-autoclaved lightweight samples prepared without silica flour. The following composites were tested: OPC/FCSs, OPC/FCSs/MGFs, OPC/FCSs/XSBR, and OPC/FCSs/MGFs/XSBR. The formulas of these systems were adapted from a 25% P/C ratio composite without silica flour. Therefore, the W/C ratios were as follows: 0.54 for OPC/FCSs, 0.56 for OPC/FCSs/MGFs, 0.32 for OPC/FCSs/XSBR, and 0.35 for OPC/FCSs/MGFs/XSBR. The resulting fracture toughness data (Figure 23) clearly demonstrated that there is a dual effect of MGFs and XSBR on toughness improvement compared against each of these additives being present separately. In fact, the toughness of samples with MGFs and XSBR alone was 0.42 and 0.62 N-mm/mm^3^, respectively. The combined sample of these components exhibited 0.83 N-mm/mm^3^ toughness, corresponding to ~98 and ~34% improvement over MGFs and XSBR alone, respectively. However, no experimental work was performed to obtain fundamental understanding of such dual effect. 

### 3.13. Morphological Alterations

Samples’ morphologies were studied to evaluate the effect of XSBR on microstructural development and its ability to alleviate the pozzolanic activity of FCSs. Figure 24 presents morphologies and elemental compositions of freshly fractured surfaces of 100 °C-autoclaved unmodified and XSBR (25% P/C)-modified composites. Ag was used as coating material to avoid charging of the sample surface. Carbon in the elemental composition comes from both organic hydrocarbons related to XSBR and inorganic carbonate in calcium carbonate. For unmodified composites (top), an aggressive pozzolanic activity of FCSs is visible. The hard and water-impermeable aluminosilicate shell of FCSs suffered from pozzolanic reactions with alkalis, Ca^+^, and OH^−^ from OPC, which eroded the whole shell structure through their alkali dissolution. The eroded shells are no longer active as insulating aggregates and create a porous microstructure, which weakens the composite and negatively affects its waterproofing properties. This is the reason why the water-fillable porosity was higher for the control than for the modified composites. The elemental composition of the cement matrix away from FCSs consisted of Ca, Si, C, and O as major elements and Fe, S, Al, and Mg as minor ones. The C/Ca, Al/Ca, and Si/Ca atomic ratios were 2.00, 0.28, and 0.99, respectively. Thus, the matrix seems to be constructed mainly by C-S-H, CH, and calcium carbonate. In contrast, the morphology (bottom) of the modified composite was very different. The FCSs’ shells remained intact, clearly demonstrating that XSBR adequately protected them against pozzolanic reaction-led erosion. Consequently, the FCSs were able to serve as thermal insulators in the modified composite and participate in its strengthening and maintaining its waterproofing properties as water-impermeable microsphere aggregates. The atomic ratios in the square area were as follows: 5.16 C/Ca, 0.08 Al/Ca, and 0.66 Si/Ca. Compared with the unmodified one, there were two major differences: a very high quantity of C and considerably lower Al and Si concentrations, while the quantity of Ca was the same as in non-modified cement. This suggests that the matrix was more likely covered by type 1 complexes rather than C-S-H, CH, and CaCO_3_. 

Figure 25 shows morphologies and compositions of an FCS that underwent pozzolanic activities and a non-reacted FCS. For the unmodified composite, the morphology (top) of FCS particles depicts an undesirable pozzolanic activity of FCS shell surface, leading to the precipitation of pozzolanic reaction products. As is evident from 0.86 Si/Ca and 0.14 Al/Ca ratios, since FCS shells are constituted of mullite (aluminosilicate) and silica, this reaction products may be due to C-S-H and calcium aluminate silicate (CaO.Al_2_O.SiO_2_.H_2_O, C-A-S-H) formed in pozzolanic reactions of silica and aluminosilicate in FCSs with Ca^2+^, OH^−^ alkalis dissociated from OPC. In contrast, in the modified composite (bottom), FCSs had a smooth surface without any precipitation of reaction products. The elemental composition analysis gave 5.87 C/Ca, 0.71 Al/Ca, and 1.77 Si/Ca ratios. This C/Ca ratio was nearly 3-fold higher than in the unmodified sample. Furthermore, the values of Al/Ca and Si/Ca ratios were ~5- and ~2-fold higher compared to those in the unmodified composite. Hence, since Al and Si come from underlying FCS surfaces, a possible explanation is that this smooth surface of FCSs was covered with type 1 complexes because of the combination of very high C relevant to organic hydrocarbon and low Ca. Such coverage of type 1 complexes acted to protect the FCS shell surfaces from pozzolanic reaction-led erosion. 

### 3.14. Corrosion Mitigation of CS

One of the most important functions of a cement sheath in a geothermal well is CS casing protection against corrosion by highly corrosive geothermal fluids. The ability of cement composites to protect CS from brine-caused corrosion was assessed for the 100 °C-autoclaved composites made with 0, 5, 15, 25% P/C ratios before and after TS. For pre-TS-test samples, Figure 26 shows the cathodic–anodic polarization curves of the voltage, V, versus current, A, and the changes in corrosion potential (E_corr_.) and corrosion current (I_corr_.) as well as the onset cathodic current (OCC) at the beginning of the curve. Each curve for various P/C ratios is an average of three measurements at three different locations. With the unmodified composite denoted as 0% P/C ratio, the values of the E_corr_. and I_corr_. were obtained from extrapolation of the linear parts of the curves with Tafel fit as shown in the figure. The E_corr_ value directly reflects the extent of adhesion and coverage of composite on CS surfaces; namely, better adhesion and void-free coverage give more positive E_corr_. values. As to the I_corr_, a low I_corr_ value means lesser cathodic reaction of oxygen reduction, 2H_2_O + O_2_ + 4e^−^ → 4OH^−^, at the cathodic corrosion site of CS. Thus, both an upgraded waterproofing and water repellency of composite layer and surface will reduce water permeability through the composite layer, thereby inhibiting the cathodic reaction.

Based upon the cathodic corrosion-abating concepts described above, the comparison between the curves’ features of unmodified and modified composites demonstrates a 2.9 × 10^−4^ A OCC value of the 0% P/C ratio shifting to a lower current with increasing P/C ratios. In fact, 6.9 × 10^−5^ A of the 25% P/C ratio was equivalent to a 75% lower current value compared to the unmodified one, underscoring that higher XSBR content offered better protection from cathodic corrosion. Furthermore, the E_corr_. value of CS under the unmodified composite layer shifted to positive potential values with increasing P/C ratios suggesting that the adhesion and coverage of composites on CS surfaces also improved with higher XSBR content. 

For post-TS test samples (Figure 27), the Tafel curves considerably differed from the pre-TS samples for the 15 and 25% P/C ratios. Compared with the OCC values of the 0% and 5% PC ratios, the 15 and 25% P/C ratios’ OCC values shifted strikingly to 1.3 × 10^−9^ and 7.9 × 10^−10^ A, respectively, from 4.9 × 10^−4^ A for the 0% P/C and 1.2 × 10^−4^ A for the 5% P/C. As aforementioned for the water-fillable porosity and surface hydrophobicity results, the post-TS test 15 and 25% P/C ratio samples displayed a better waterproofing and water repellency than before TS. Thus, this is the reason why the cathodic reaction was strongly inhibited. The E_corr_. value for the 15 and 25% P/C ratios shifted to the positive potential region of 0.2 to 0.3 V from a negative potential, ranging from −0.3 to −0.2 V, for the 0 and 5% P/C ratios. The TS appeared to improve the adhesion and coverage of the XSBR-modified composites on CS surfaces. 

Figure 28 and Figure 29 summarize the I_corr_. and E_corr_. results for unmodified and modified composite coatings before and after TS. The I_corr_. and E_corr_. values are averages obtained from three different locations, except for the 15 and 25% P/C ratios after the TS tests. For these samples the protection in two out of three tested locations was so strong that the potential and the current needed for the measurements were over the capacity of the instrument which could only be solved if the coating thickness would be decreased. 

Figure 30 compares CS corrosion rates in millimeter per year (mm/year) computed from Tafel fit results for unmodified and modified composite coatings before and after TS. The average thicknesses of the composite coating layers over the underlying CS plates for samples before and after TS are also included in this figure. The coating thicknesses in millimeter (mm) of the 0, 5, 15, and 25% P/C ratio composites for the pre- and post-TS test samples were in the range of 0.27–0.41, 0.18–0.29, 0.19–0.24, and 0.45–0.46, respectively. For pre-TS test samples, the corrosion rate of the 0% P/C ratio sample was 0.71 mm/year. This corrosion rate reduced by 34% for the 5% P/C sample. Further corrosion rate reduction was observed for the 15% and 25% P/C ratio coatings corresponding to 48% and 87% corrosion abatement with 3.6 × 10^−1^ and 9.0 × 10^−2^ mm/year corrosion, respectively. For post-TS test samples, the corrosion rate of the 0% P/C sample slightly increased to 0.75 mm/year compared with that of the pre-TS sample. In contrast, all modified composite coatings offered a significantly improved corrosion protection of CS, manifesting superior cathodic corrosion protection due to the minimized cathodic reactions, and advanced adhesion and coverage of coating on CS surfaces. In particular, the 15% and 25% P/C ratio coatings incredibly reduced the corrosion rate to 3.5 × 10^−6^ and 1.5 × 10^−6^ mm/year, respectively. Since TS led to the formation of type 2 multiple complexes, these complexes enhanced the hydrophobicity and minimized the water-fillable porosity. The type 2 complexes may also assist in improving adhesion and coverage of a composite on CS, playing a pivotal role in remarkably reducing CS corrosion rates.

Regarding the coating thickness of samples before and after TS, despite a lower average thickness of 0.212 mm of the 15% P/C ratio composite compared with the higher average thickness of 0.454 mm of the 25% P/C ratio sample, there was no significant difference in the corrosion rate reduction. Thus, in this modified composite systems, a coating thickness in the range of 0.212 to 0.454 mm did not affect any corrosion abating performance. 

It should be noted that the study did not include the evaluation of cement cohesion to the carbon steel. Despite excellent steel corrosion protection these Portland cement-based formulations may have poor steel adhesion resulting in debonded cement slurry under the shock conditions. Whether this is the case and whether the corrosion protection remains present to some extend due to the polymer–steel interactions must be further verified to make informed conclusions about the cement–metal interface durability and long-term steel corrosion protection under the conditions of storage wells.

## 4. Conclusions

XSBR latex, a terpolymer of polystyrene (PS), polybutadiene (PB), and polyacrylic acid (PAA), was evaluated as a potential additive for designing hydrophobic thermal insulating, thermal shock-resistant class G well cement (OPC) composites containing fly ash cenospheres (FCSs) as thermal insulator and micro-E-glass fibers (MGFs) as reinforcement at a thermal gradient of 150 °C at temperatures between 100 °C and 175 °C, for reservoir thermal energy storage systems. 

Under a hydrothermal environment at 100 °C, functional carboxyl groups (acidic) within PAA reacted with calcium ions liberated from hydrolysis of OPC to produce XSBR with Ca^2+^-coordinated PAA complexes (type 1). Elevating the hydrothermal temperature to 175 °C brought about hydrothermal degradation/oxidation cleavage of diene alkenes in PB, forming isolated carboxyl-terminated PB, and non-isolated carboxyl-end PS and -end PAA. Such alterations of XSBR terpolymer produced additional functional carboxyl groups that formed multiple Ca^2+^-coordinated complexes with calcium ions from cement (type 2). These complexes not only sequestrated Ca^2+^ ions, but also adsorbed on the hydrated OPC. Furthermore, these type 1 and 2 structures withstood hydrothermal temperatures of up to ~250 °C. 

Importantly, XSBR derivatives formed in cement slurry had the following effect on the OPC/silica blend: (1) eliminating OPC carbonation; (2) controlling OPC hydration reactions; (3) enhancing the extent of hydrophobicity; (4) decreasing water-fillable porosity; (5) reducing the blend’s thermal conductivity for water-saturated composites; (6) minimizing the loss of compressive strength, Young’s modulus as stiffness, and compressive fracture toughness in TS tests; (7) abating pozzolanic degradation of FCSs; and (8) improving corrosion protection of CS surfaces.

Moreover, XSBR modifications provided great workability and pumpability of cement slurries due to the ability of XSBR latex to act as a plasticizer.

Adhesion of the XSBR-modified cement to the carbon steel was not evaluated in this study and needs to be further investigated. 

The information described above indicates that XSBR-modified OPC has a potential for use as hydrophobic thermally insulating, thermal-shock-and-corrosion resistant well composites in reservoir thermal energy storage systems up to 175 °C. 

## Figures and Tables

**Figure 1 materials-16-05792-f001:**
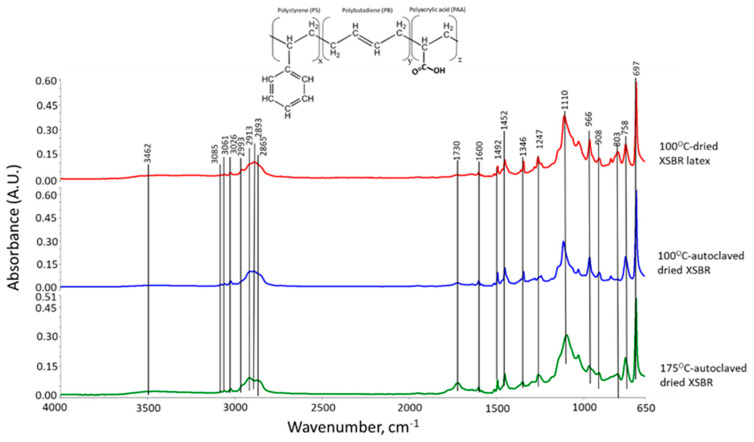
ATR-FTIR absorption spectra for 100 °C-dried XSBR latex, and 100 °C- and 175 °C-autoclaved dried XSBR samples.

**Figure 2 materials-16-05792-f002:**
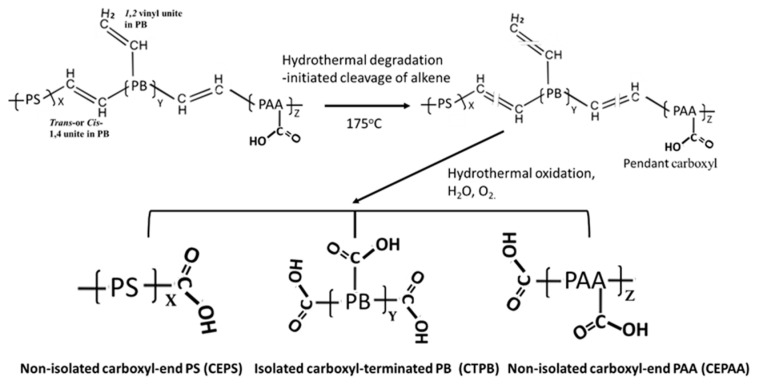
Hydrothermal degradation and oxidation pathways of PB *trans-* and *cis*-1,4, and 1,2 vinyl units to form isolated carboxyl-terminated PB (CTPB), and non-isolated carboxyl-ended PS (CEPS) and carboxyl-ended PAA (CEPAA) as oxidation derivatives.

**Figure 3 materials-16-05792-f003:**
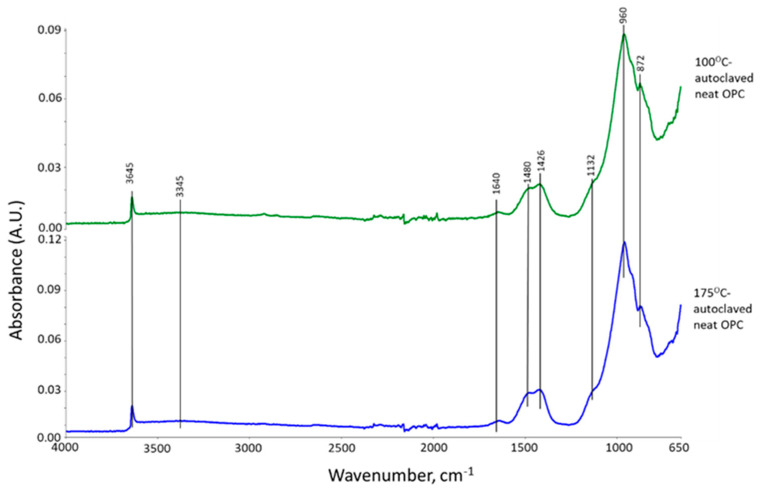
ATR-FTIR spectra for 100 °C- and 175 °C-autoclaved neat OPC.

**Figure 4 materials-16-05792-f004:**
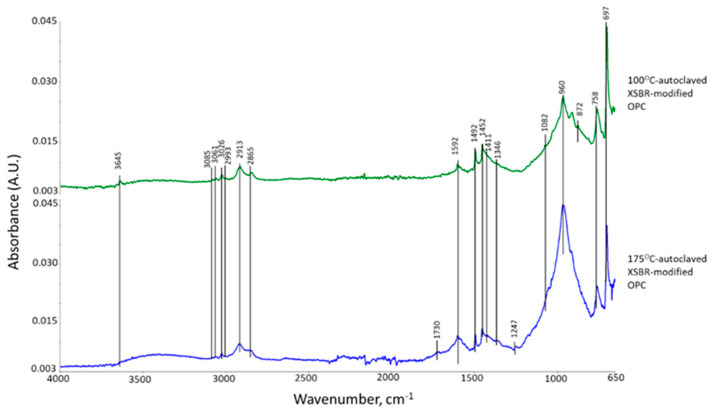
ATR-FTIR spectra of XSBR-modified neat OPC after being autoclaved at 100 °C and 175 °C.

**Figure 5 materials-16-05792-f005:**
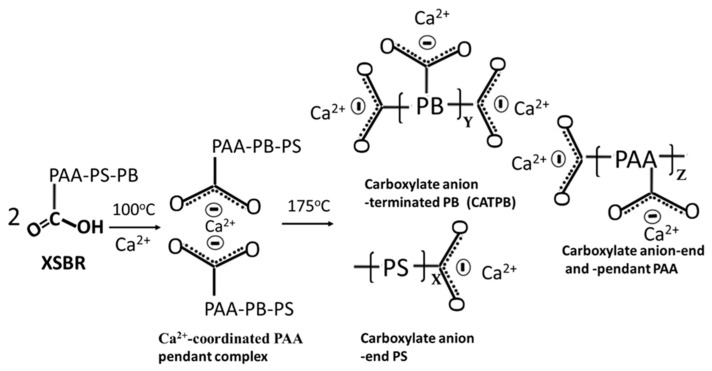
Illustration of XSBR molecular transformations in acid–base reactions between carboxyl group (proton donor acid) and Ca^2+^ 2OH^−^ (proton acceptor base) at hydrothermal temperatures of 100 °C and 175 °C.

**Figure 6 materials-16-05792-f006:**
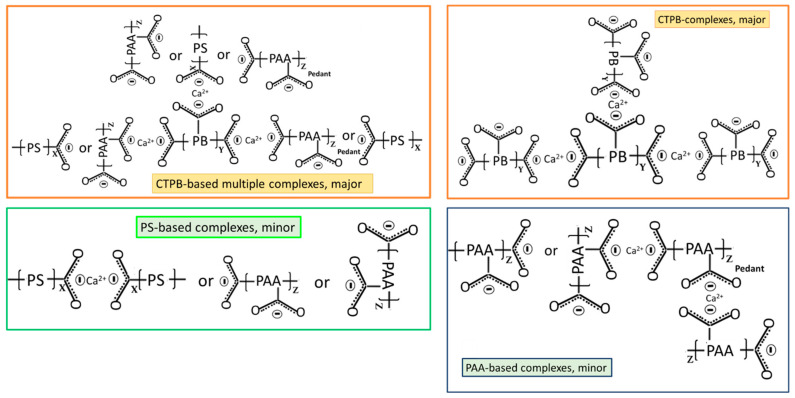
Illustration of Ca^2+^-coordinated chelate complex configurations derived from PB-hydrothermal degradation/oxidation at 175 °C.

**Figure 7 materials-16-05792-f007:**
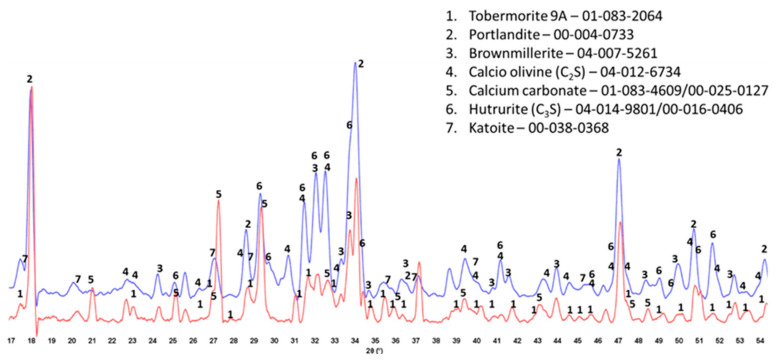
XRD patterns of crystalline phases of 175 °C-autoclaved XSBR-modified (blue) and neat OPC (red).

**Figure 8 materials-16-05792-f008:**
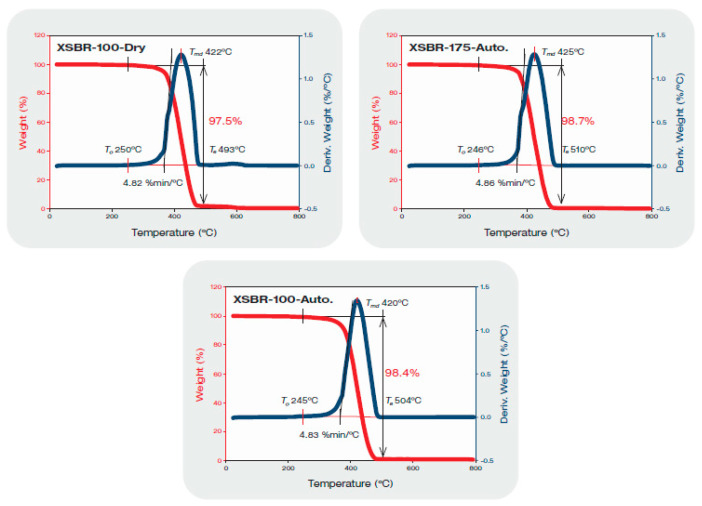
TGA-DTG thermal analyses of 100 °C-dried XSBR, and 100 °C- and 175 °C-autoclaved XSBR samples.

**Figure 9 materials-16-05792-f009:**
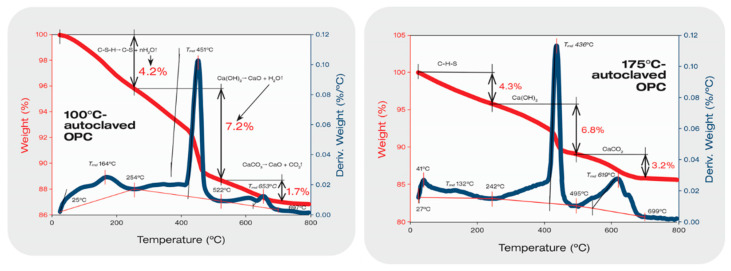
TGA-DTG thermal analyses of 100 °C- and 175 °C-autoclaved neat OPC samples.

**Figure 10 materials-16-05792-f010:**
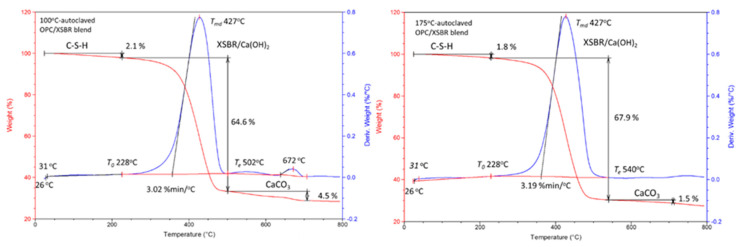
TGA-DTG thermal analyses of 100 °C- and 175 °C-autoclaved XSBR-modified OPC samples.

**Figure 11 materials-16-05792-f011:**
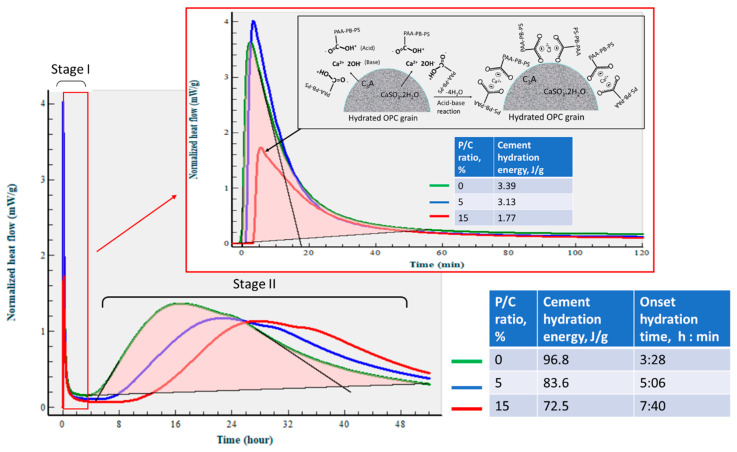
Normalized heat flow curves and hydration reaction energies of OPC slurries made with 0, 5, and 15% P/C ratios at 25 °C. Illustration of suppression of ettringite formation by XSBR containing PAA-pendant chelate complexes adsorbed on cement grain surfaces.

**Figure 12 materials-16-05792-f012:**
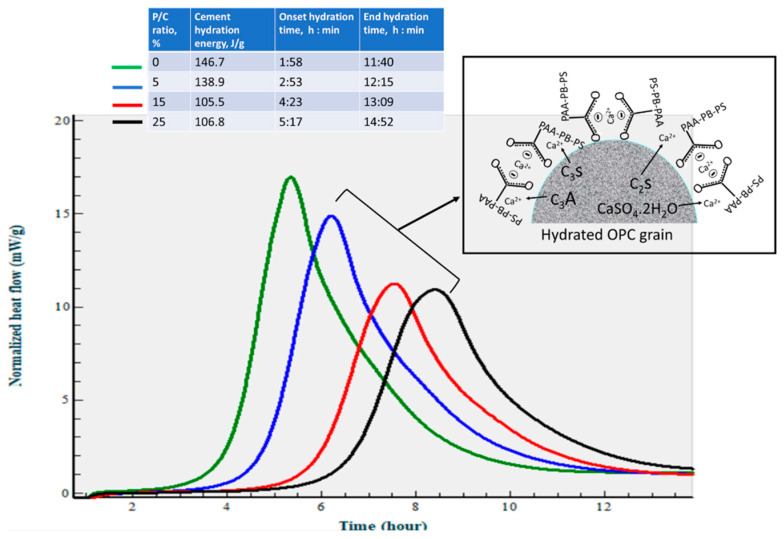
Normalized heat flow curves and hydration reaction energies of OPC slurries made with 0, 5, 15, and 25% P/C ratios at 85 °C.

**Figure 13 materials-16-05792-f013:**
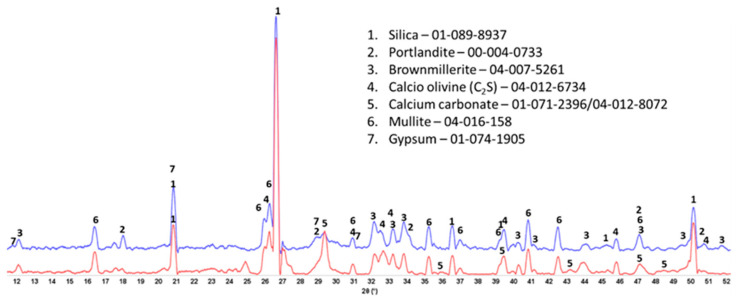
XRD patterns of 100 °C-autoclaved OPC composites modified (blue) and unmodified (red) with XSBR.

**Figure 14 materials-16-05792-f014:**
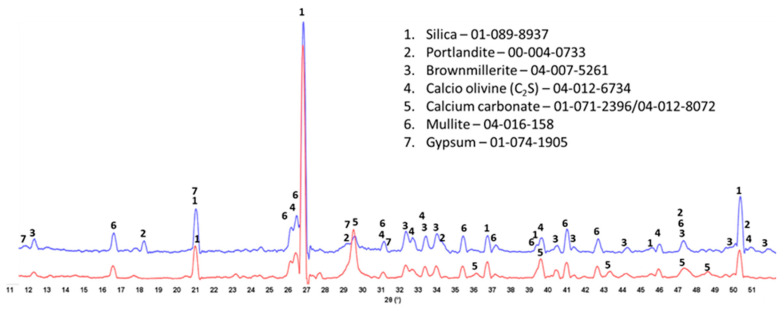
XRD patterns of 175 °C-autoclaved OPC composites modified (blue) and unmodified (red) with XSBR.

**Figure 15 materials-16-05792-f015:**
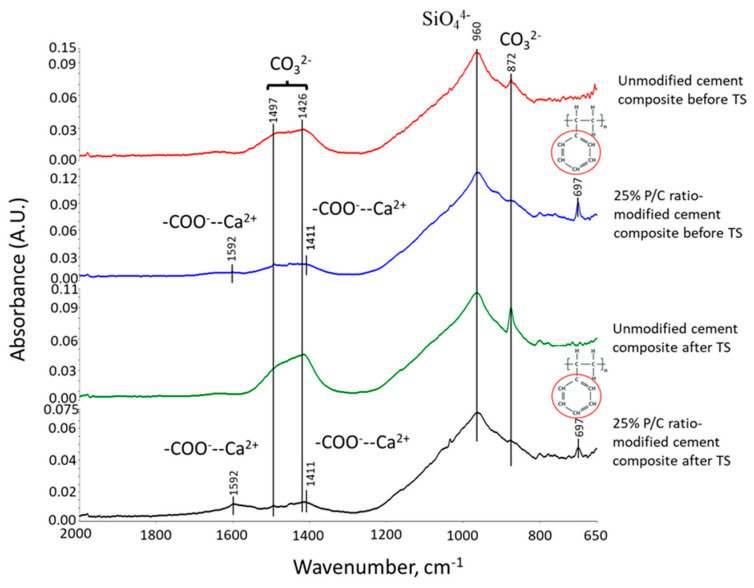
ATR-FTIR spectra of 25% P/C ratio-modified and unmodified OPC composites before and after TS tests.

**Figure 16 materials-16-05792-f016:**
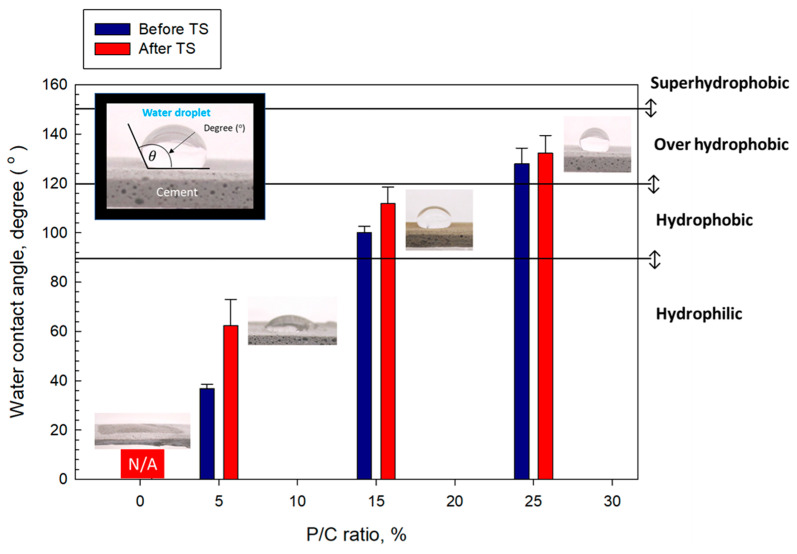
Wetting behavior and water droplet contact angle for 5, 15, and 25% P/C ratio-modified and unmodified OPC composite surfaces before and after TS tests.

**Figure 17 materials-16-05792-f017:**
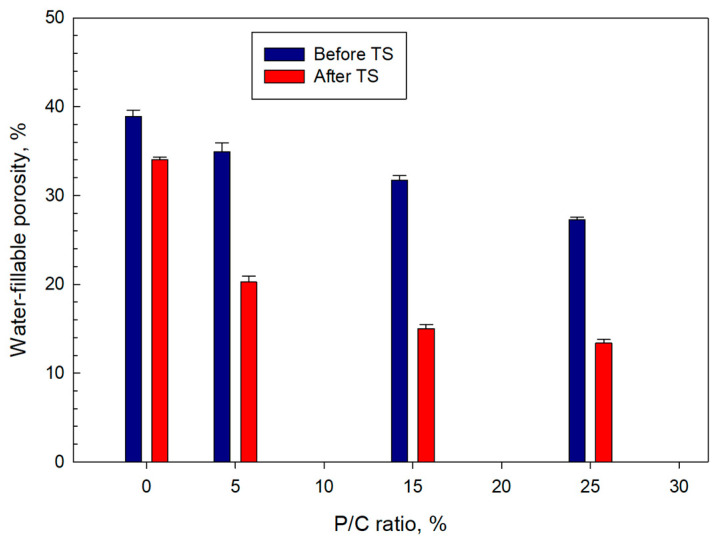
Water-fillable porosity of 5, 15, and 25% P/C ratio-modified and unmodified composites before and after TS tests.

**Figure 18 materials-16-05792-f018:**
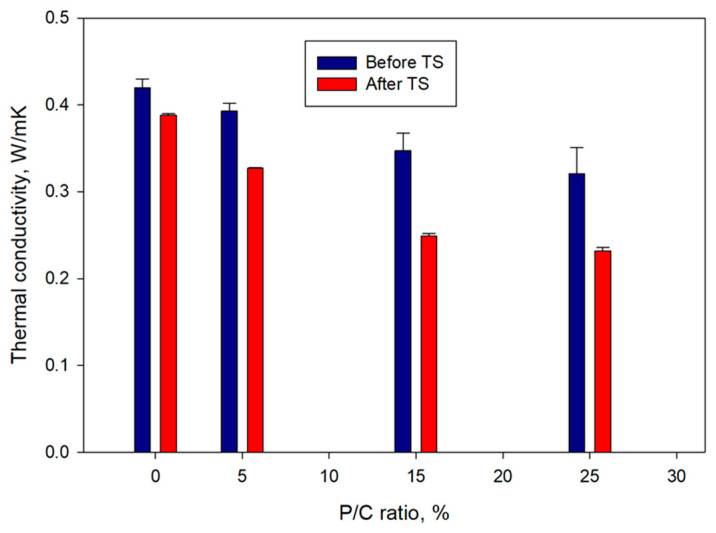
Thermal conductivity of XSBR-modified and unmodified OPC composites before and after TS tests.

**Figure 19 materials-16-05792-f019:**
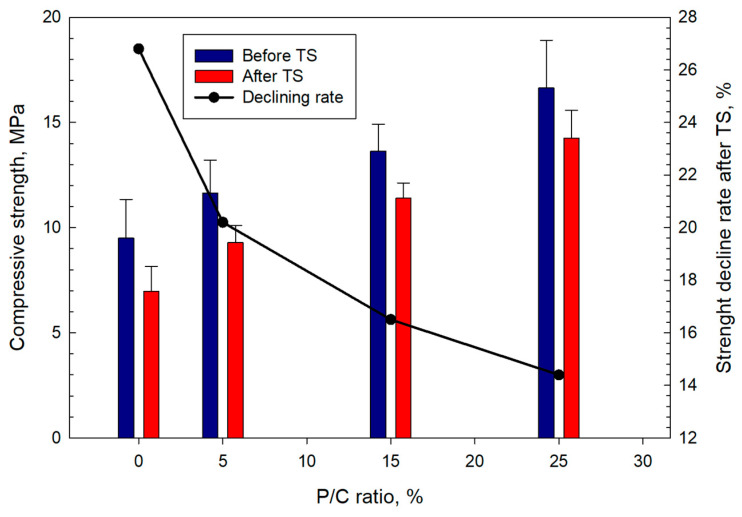
Changes in compressive strength as a function of P/C ratio, and strength decline rate after TS tests.

**Figure 20 materials-16-05792-f020:**
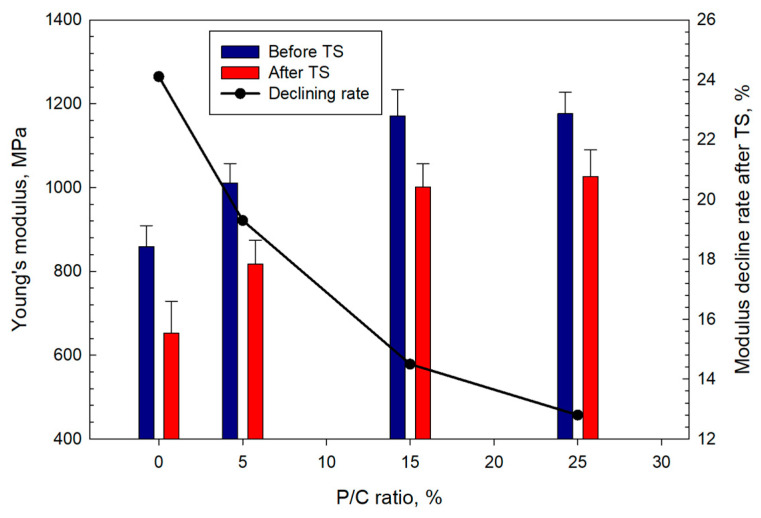
Changes in Young’s modulus as a function of P/C ratio, and the modulus decline rate after TS tests.

**Figure 21 materials-16-05792-f021:**
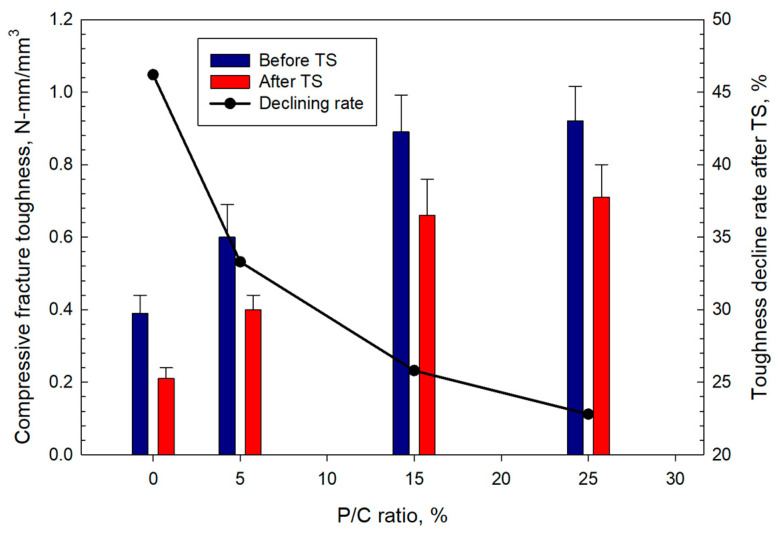
Changes in compressive fracture toughness as a function of P/C ratio, and toughness decline rate after TS tests.

**Figure 22 materials-16-05792-f022:**
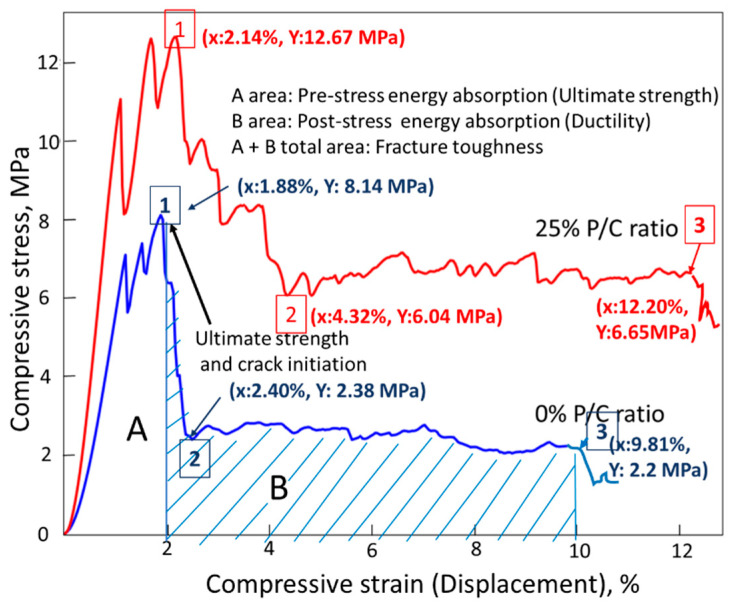
Comparison of compressive fracture toughness on the compressive stress–strain curves for 0% and 25% P/C ratio OPC composites after TS tests.

**Figure 23 materials-16-05792-f023:**
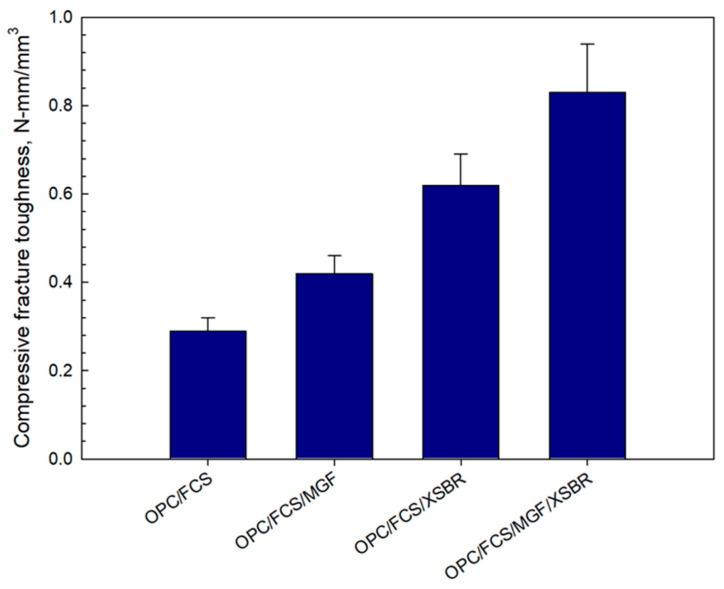
Comparison of compressive fracture toughness for OPC/FCSs, OPC/FCSs/MGFs, OPC/FCSs/XSBR, and OPC/FCSs/MGFs/XSBR combination systems.

**Figure 24 materials-16-05792-f024:**
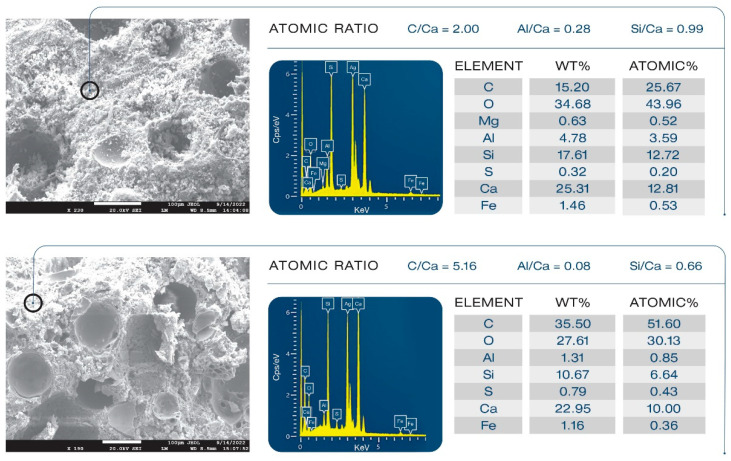
Comparison of morphological features between unmodified (**top**) and XSBR-modified (**bottom**) OPC composites.

**Figure 25 materials-16-05792-f025:**
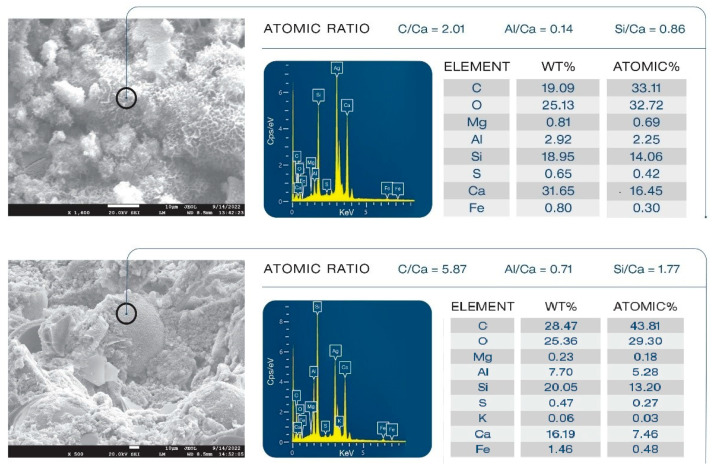
Pozzolanic reaction products of FCSs in unmodified composite (**top**) and non-reacted FCSs in XSBR-modified OPC composite (**bottom**).

**Figure 26 materials-16-05792-f026:**
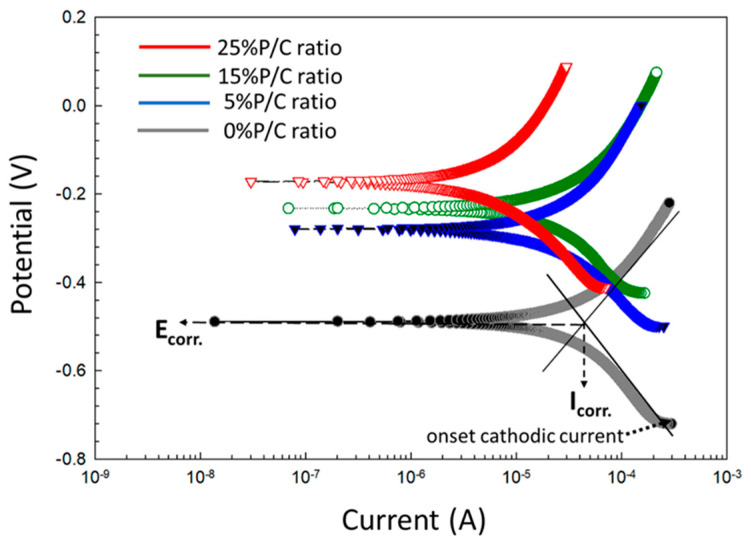
DC-electrochemical cathodic–anodic polarization curves for 0, 5, 15, and 25% P/C ratio cement composite-coated CS surfaces and Tafel fit diagram before TS test.

**Figure 27 materials-16-05792-f027:**
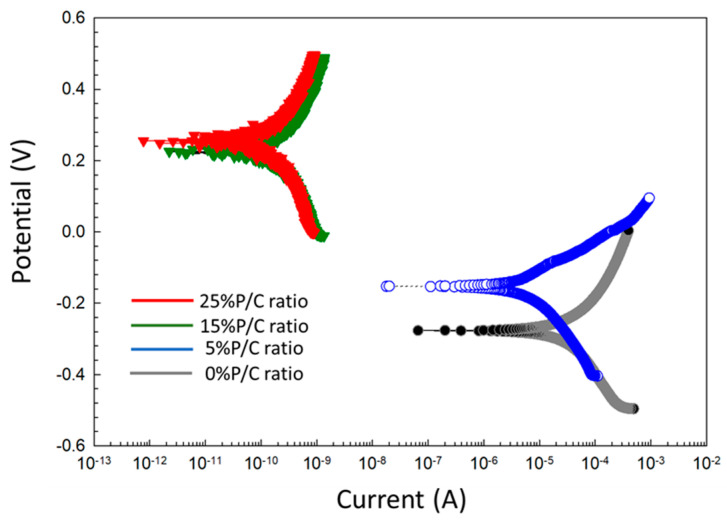
DC-electrochemical cathodic–anodic polarization curves for 0, 5, 15, and 25% P/C ratio cement composite-coated CS surfaces after TS tests.

**Figure 28 materials-16-05792-f028:**
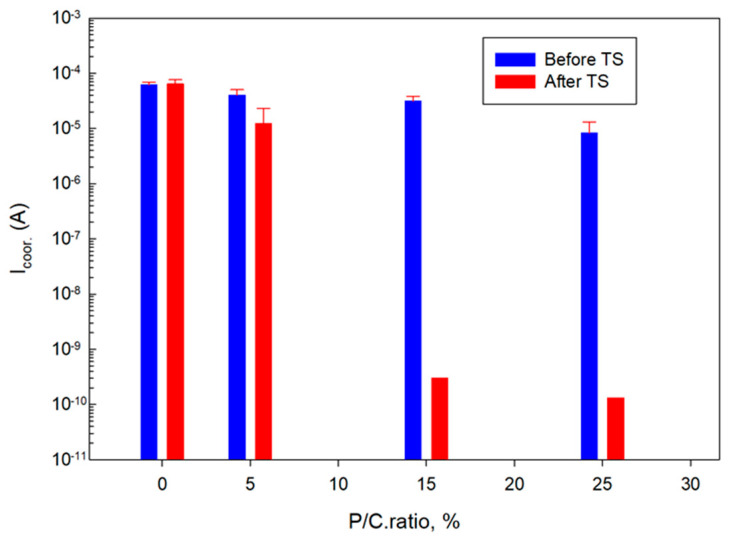
Corrosion current as a function of P/C ratio before and after TS tests.

**Figure 29 materials-16-05792-f029:**
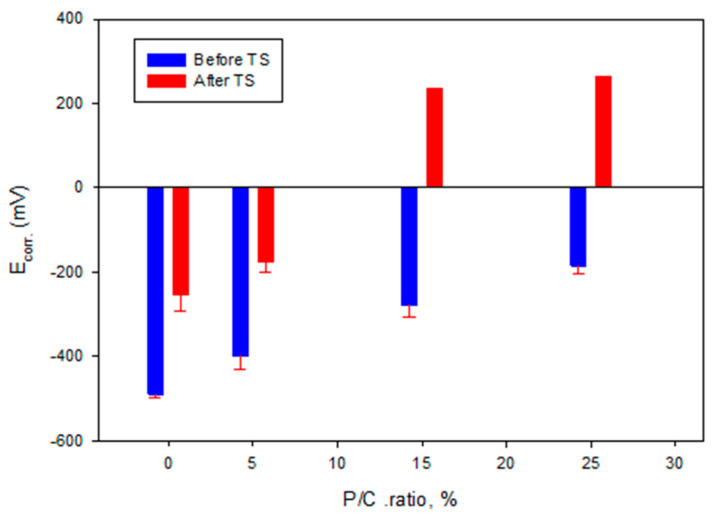
Corrosion potential as a function of P/C ratio before and after TS tests.

**Figure 30 materials-16-05792-f030:**
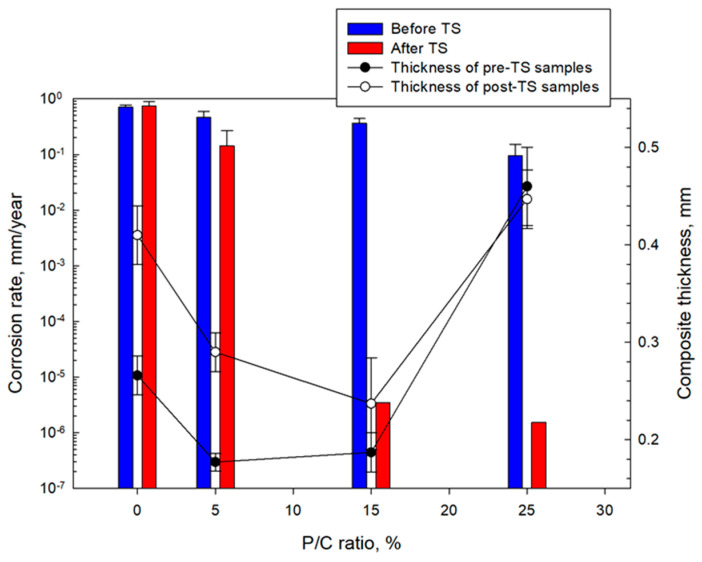
Corrosion rate of CS and thickness of coating layers for 0, 5, 15, and 25% P/C ratio composite coatings before and after TS tests.

**Table 1 materials-16-05792-t001:** Oxide compositions of starting materials.

Component	Oxide Composition, wt.%								
	Al_2_O_3_	CaO	SiO_2_	Fe_2_O_3_	Na_2_O	K_2_O	TiO_2_	MgO	SO_3_
Class G cement	3.0	67.6	18.4	3.9	0.3	1.3	-	-	5.5
Silica flour	-	-	100	-	-	-	-	-	-
FCSs	35.0	2.7	50.1	7.1	0.30	3.1	1.6	-	-
E-type MGFs	11.4	28.6	55.0	0.9	0.6	-	0.7	2.8	-

**Table 2 materials-16-05792-t002:** Absorbance height ratios (ΔA_1730 cm−1_/ ΔA_1600 cm−1_) between -COOH at 1730 cm^−1^ and PB C=C at 1600 cm^−1^, and changes in height of ΔA_966 cm−1_ of PB *trans*-1,4 and ΔA_908 cm−1_ of PB 1,2 vinyl at 100 °C and 175 °C.

	ΔA_1730 cm−1_	ΔA_1600 cm−1_	ΔA_1730 cm−1_/ΔA_1600 cm−1_	ΔA_966 cm−1_	ΔA_908 cm−1_
100 °C-dried XSBR	0.008	0.017	0.471	0.125	0.034
100 °C-autoclaved dried XSBR	0.015	0.031	0.484	0.125	0.038
175 °C-autoclaved dried XSBR	0.042	0.02	2.1	0.03	0.017

**Table 3 materials-16-05792-t003:** Properties of XSBR-modified and unmodified OPC composite slurries at ambient temperature.

Property	P/C Ratio, %			
	0	5	15	25
Water-to-composite ratio (W/C)	0.51	0.45	0.37	0.31
Density, d/cm^3^	1.26	1.11	1.06	1.04
Slump size, mm	60	64	72	75

## Data Availability

The data presented in this study are available on request from the corresponding author.

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
