# Peer review of "Hydrophobic, Thermal Shock-and-Corrosion-Resistant XSBR Latex-Modified Lightweight Class G Cement Composites in Geothermal Well Energy Storage Systems"

_materials, 2023, doi:10.3390/ma16175792_

Round 1
Reviewer 1 Report
Manuscript ID: Materials-2509898
Title: Hydrophobic, thermal shock-and corrosion-resistant XSBR latex-modified lightweight Class G cement composites in geothermal well energy storage system
Journal: Materials
Type: Article
Comments to authors:
In this paper the factors governing the potential of XSBR (cross-linked styrene butadiene rubber) latex-modified Class G cement composites with FCS (Fly ash cenospheres) as a thermal insulator and MGF (micro-glass fiber) as a reinforcement for applications in RTES (Reservoir thermal energy storage) systems were investigated. The detailed investigation include tests on hydrothermal stability and reactions, phase compositions, slurry properties, thermal conductivity, mechanical properties, morphological alterations and corrosion resistance.
The paper is well written and includes detailed discussion on the tested properties and their results.
1. The introduction section should include more related previous studies. The authors have given reference to their own previous studies; however, it needs to be expanded to include works by others on similar areas to complete the literature review and the individual standing of the present study.
2. What are the limitations of the present study? Please mention them in the manuscript in an appropriate section.
3. Conclusions section should be refined and briefly presented by including bullet point for an inference/conclusion.
4. The authors should add a separate heading for the future recommendations based on the present study.
Author Response
Reviewer-1
Comments to authors:
In this paper the factors governing the potential of XSBR (cross-linked styrene butadiene rubber) latex-modified Class G cement composites with FCS (Fly ash cenospheres) as a thermal insulator and MGF (micro-glass fiber) as a reinforcement for applications in RTES (Reservoir thermal energy storage) systems were investigated. The detailed investigation include tests on hydrothermal stability and reactions, phase compositions, slurry properties, thermal conductivity, mechanical properties, morphological alterations and corrosion resistance.
The paper is well written and includes detailed discussion on the tested properties and their results.
- The introduction section should include more related previous studies. The authors have given reference to their own previous studies; however, it needs to be expanded to include works by others on similar areas to complete the literature review and the individual standing of the present study.
Seven additional references on latex applications in underground wells were added to the introduction. For the thermally insulating cement references, additional references can be found in the first two papers of the series, with the current one being the last.
- What are the limitations of the present study? Please mention them in the manuscript in an appropriate section.
The discussion on limitations of the study was added to the corrosion part of the paper and mentioned in the conclusions.
- Conclusions section should be refined and briefly presented by including bullet point for an inference/conclusion.
The conclusion section was modified.
- The authors should add a separate heading for the future recommendations based on the present study.
Potential of the developed technology is stated in the conclusions. We prefer not to make a separate section on this subject.
Submission Date
30 June 2023
Date of this review
18 Jul 2023 09:04:08

Reviewer 2 Report
In the Reviewer opinion the research paper entitled “ Hydrophobic, thermal shock-and corrosion-resistant XSBR latex-modified lightweight Class G cement composites in geothermal well energy storage system” is very good.
In this study authors assessed the ability of polystyrene (PS)-polybutadiene (PB)-polyacrylic acid (PAA) terpolymer (cross-linked styrene-butadiene rubber, XSBR) latex to improve thermal insulating properties and thermal shock (TS) resistance of class G Ordinary Portland Cement (OPC) and fly ash cenospheres (FCS) composites in the temperature range of 100o-175oC. The composites autoclaved at 100oC were subjected to 3 cycles of TS (one cycle: 175oC heat → 25oC water quenching). In hydrothermal and thermal (TS) environments at elevated temperatures in cement slurries the XSBR latex formed acrylic calcium complexes through ac-id-base reactions, and the number of such complexes increased at higher temperatures due to the XSBR degradation with formation of additional acrylic groups.
Some comments which greatly enhance the understanding of the paper and its value are presented below. Specific issues that require further consideration are:
- The title of the manuscript is matched to its content.
- The Introduction generally covers the cases.
- The methodology was clearly presented.
- In the Reviewer’s opinion, the current state of knowledge relating to the manuscript topic has been presented.
- Experimental program and results looks interesting and was clearly presented.
- In the Reviewer’s opinion, the bibliography, comprising 46 references, is representative.
- An analysis of the manuscript content and the References shows that the manuscript under review constitutes a summary of the Author(s) achievements in the field.
- In the Reviewer’s opinion the manuscript is well written, and it should be published in the journal.
Reviewer 3 Report
It is an intriguing work and the manuscript has demonstrated a comprehensive scope of research work with well-analysed results. However, there are some concerns related to certain experiments. I recommend the authors to address these comments during their revision.
1. Line 156: Please clarify whether the term "C" refers to the composite of dry blend.
2. In sections 2.2 and 2.3, including a table of mixture composition and a schematic diagram illustrating the testing process, and its specimens, would greatly aid the reader's understanding.
3. Line 181: Could you explain why the miniature slump cone deviates from the standard size proportions, such as 1 (top diameter): 2 (bottom diameter): 3 (height)?
4. Line 187: Please include the immersion duration in 25 C water.
5. Line 203: The term "water repellency" might not be suitable in this context, since the paragraph pertains to porosity measurement.
6. Line 249: Why "Hydrothermal stability" was used to recall the specimen tested using FTIR and "Thermal stability" for TGA test ? As both used similar autoclaved specimens.
7. Figure 19, 20, and 21: The differences between "Before-After TS" specimens seem consistent at similar rate from one mixture to another (based on the primary data). How the declining rate was calculated to obtain up to 40% differences?
8. Figure 22: There is a flaw in this analysis. The specimen was treated similar to the steel material. Unlike steel, there is no "necking" phase in cementitious material. Since there is no obvious evidence in the prolonged deformation after the specimen has reached the yield stress in the manuscript, this ductility should not be referred as the proposed "B" area.
9. Line 738: If pozzolanic reaction is undesirable, the experiment should also be carried out by investigating a mixture that does not contain fly ash and additional silica in the ingredients. Since SEM images only focused at the micro spots in the specimen, the display of eroded and unreacted fly ash particle might be inaccurate if the specimen was not properly extracted.
Round 2
Reviewer 3 Report
The authors have attentively addressed the comments. Based on my assessment, I recommend the acceptance of the article for publication.